# RandNet-Parareal: a time-parallel PDE solver using Random Neural Networks

**Guglielmo Gattiglio**
Department of Statistics
University of Warwick
Coventry, CV4 7AL, UK
Guglielmo.Gattiglio@warwick.ac.uk

**Lyudmila Grigoryeva**[*]
Faculty of Mathematics and Statistics
University of St. Gallen
Rosenbergstrasse 20, CH-9000 St. Gallen, Switzerland
Lyudmila.Grigoryeva@unisg.ch

**Massimiliano Tamborrino**
Department of Statistics
University of Warwick
Coventry, CV4 7AL, UK
Massimiliano.Tamborrino@warwick.ac.uk

## Abstract

Parallel-in-time (PinT) techniques have been proposed to solve systems of time-dependent differential equations by parallelizing the temporal domain. Among them, Parareal computes the solution sequentially using an inaccurate (fast) solver, and then "corrects" it using an accurate (slow) integrator that runs in parallel across temporal subintervals. This work introduces RandNet-Parareal, a novel method to learn the discrepancy between the coarse and fine solutions using random neural networks (RandNets). RandNet-Parareal achieves speed gains up to x125 and x22 compared to the fine solver run serially and Parareal, respectively. Beyond theoretical guarantees of RandNets as universal approximators, these models are quick to train, allowing the PinT solution of partial differential equations on a spatial mesh of up to $10^5$ points with minimal overhead, dramatically increasing the scalability of existing PinT approaches. RandNet-Parareal's numerical performance is illustrated on systems of real-world significance, such as the viscous Burgers' equation, the Diffusion-Reaction equation, the two- and three-dimensional Brusselator, and the shallow water equation.

## 1 Introduction

Parallel-in-time (PinT) methods have been used to overcome the saturation of well-established spatial parallelism approaches for solving (prohibitively expensive) initial value problems (IVPs) for ordinary and partial differential equations (ODEs and PDEs), described by systems of $d \in \mathbb{N}$ ODEs (and

---

[*]Honorary Associate Professor, Department of Statistics, University of Warwick, Coventry, CV4 7AL, UK. (Lyudmila.Grigoryeva@warwick.ac.uk)

38th Conference on Neural Information Processing Systems (NeurIPS 2024).

similarly for PDEs)

$$\frac{d\boldsymbol{u}}{dt} = h(\boldsymbol{u}(t), t) \quad \text{on } t \in [t_0, t_N], \ \text{ with } \boldsymbol{u}(t_0) = \boldsymbol{u}^0, \ N \in \mathbb{N}, \tag{1}$$

where $h : \mathbb{R}^d \times [t_0, t_N] \to \mathbb{R}^d$ is a smooth multivariate function, $\boldsymbol{u} : [t_0, t_N] \to \mathbb{R}^d$ is the time dependent column vector solution, and $\boldsymbol{u}^0 \in \mathbb{R}^d$ is the initial value at $t_0$. PinT schemes are particularly important when the sequential application of an accurate numerical integrator $\mathscr{F}$ over $[t_0, t_N]$ is infeasible in a reasonable wallclock time. There are three general approaches for PinT computation: parallel across-the-problem, parallel-across-the-step, and parallel-across-the-method. In [17, 55], another classification is provided: multiple shooting, methods based on waveform relaxation and domain decomposition, multigrid approaches, and direct time-parallel methods. Parallel-across-the-step methods, in which solutions at multiple time-grid points are computed simultaneously, include Parareal (approximation of the derivative in the shooting method) [45], Parallel Full Approximation Scheme in Space and Time (PFASST) (multigrid method) [13, 50], and Multigrid Reduction in Time (MGRIT) [14, 16] methods (see [19] for details). Among them, Parareal [45] has garnered popularity, with extensive theoretical analyses, improved versions, and empirical applications [17, 55]. This is due to its non-intrusive nature which allows seamless integration with arbitrary temporal and spatial discretizations, and to its successful performance across diverse fields, such as plasma physics [64, 66, 67], finance [4, 56], and weather modeling [59, 60]. Limited theoretical results are available for MGRIT and PFASST, with a few extensions and empirical applications. Interestingly, combined analyses have shown equivalences between Parareal and MGRIT, and connections between MGRIT and PFASST. In Parareal, a coarse and fast solver $\mathscr{G}$ is run sequentially to obtain a first approximation of the solution, which is then corrected by running a fine (accurate) but slow integrator $\mathscr{F}$ in parallel across $N$ temporal subintervals. This procedure is then iterated until a convergence criterion is met after $k \leq N$ iterations, leading to a speed-up compared to running $\mathscr{F}$ sequentially over the entire time interval. A recent advancement, GParareal [57], improves Parareal convergence rates (measured as $k/N$) by learning the discrepancy $\mathscr{F} - \mathscr{G}$ using Gaussian Processes (GPs). This method outperforms Parareal for low-dimensional ODEs and a moderate number of computer cores $N$. However, the cubic cost (in the number of data points, roughly $kN$ at iteration $k$) of inverting the GP covariance matrix hinders its broader application. Subsequent research introduced nearest neighbors (nns) GParareal (nnGParareal) [21], enhancing GParareal's scalability properties in both $N$ and $d$ through data reduction. Significant computational gains were achieved by training the GP on a small subset of nns, resulting in an algorithm loglinear in the sample size. This allowed scaling its effectiveness up to systems with a few thousand ODEs, beyond which it loses its potential. Indeed, being based on the original GP framework, it uses a costly hyperparameter optimization procedure that requires fitting one GP per ODE dimension.

This study introduces RandNet-Parareal, a new approach using random neural networks (RandNets) to learn the discrepancy $\mathscr{F} - \mathscr{G}$. RandNets are a family of single-hidden-layer feed-forward neural networks (NNs), where hidden layer weights are randomly sampled and fixed, and only the output (or readout) layer is subject to training. Compared to standard artificial NNs, RandNets are hence much simpler to train: the input data are fed through the network, the predictions observed, and the weights of the linear output (or readout) layer are obtained as minimizers of a penalized squared loss between the NN outputs and the training targets. Since this optimization problem admits a closed-form solution, no backpropagation is required, and the issues of vanishing and exploding gradients persisting for standard fully trainable NNs are therefore avoided. The literature on the topic is rich and somewhat fragmented, and different names are used for essentially the same model. RandNets are related to Random Feature Networks [6, 49, 62, 63, 65] and Reservoir Computing [24, 26, 25, 27, 28], Random Fourier Features (RFFs) and kernel methods [41, 61, 70, 74]. Some authors use the name Extreme Learning Machines (ELMs) [34–37, 44] to refer to RandNets, while others use the term randomized or random NNs [5, 32, 39, 46, 78, 82] for the same paradigm. RandNets show excellent empirical performance, and have been used in the context of mathematical finance [22, 33, 38], mathematical physics [52], electronic circuits [69], photonic [47] and quantum systems [23, 48], random deep splitting schemes [53], scientific computing [10, 11, 79, 81], and have shown excellent empirical performance in numerous further applications. Moreover, recent work [22, 25] proves that RandNets are universal approximators within spaces of sufficiently regular functions, and provides explicit approximation error bounds, with these results generalized to a large class of Bochner spaces in [52]. These contributions show that RandNets are a reliable machine learning paradigm with provable theoretical guarantees.

In this paper, we show that endowing Parareal with RandNets-based learning of $\mathscr{F} - \mathscr{G}$, the new proposed RandNet-Parareal algorithm, leads to significantly improved scalability, convergence speed, and parallel performance with respect to nnGParareal, GParareal, and Parareal. This allows us to solve PDE systems on a fine mesh of up to $10^5$ discretization points with negligible overhead, outperforming nnGParareal by two orders of magnitude and reducing its model cost by several orders.

Here, we compare the performance of Parareal, nnGParareal, and RandNet-Parareal on five increasingly complex systems, some of which are drawn from an extensive benchmark study of time-dependent PDEs [75]. These include the one-dimensional viscous Burgers' equation, the two-dimensional Diffusion-Reaction equation, a challenging benchmark used to model biological pattern formation [76], the two- and three-dimensional Brusselator, known for its complex behavior, including oscillations, spatial patterns, and chaos, and the shallow water equations (SWEs). Derived from the compressible Navier-Stokes equations, the SWEs are a system of hyperbolic PDEs exhibiting several types of real-world significance behaviors known to challenge numerical integrators, such as sharp shock formation dynamics, sensitive dependence on initial conditions, diverse boundary conditions, and spatial heterogeneity. Example applications include of tsunamis or flooding simulations.

We intentionally chose two hyperbolic equations (Burgers' and SWE) to challenge RandNet-Parareal on systems for which Parareal is known to struggle, with slow or non-convergent behavior [2, 3, 9, 18, 72]. Previous works have developed ad-hoc coarse solvers to address Parareal's slow convergence for Burgers' [7, 40, 68, 71], and for SWE [1, 31, 54, 73]. Here, we adopt a different strategy: by leveraging the generalization capabilities of RandNets within the Parareal algorithm, we enhance the performance of standard, off-the-shelf integration methods such as Runge-Kutta, obtaining speed gains up to x125 and x22 compared to the accurate integrator $\mathscr{F}$ and Parareal, respectively. All experiments have been executed on Dell PowerEdge C6420 compute nodes each with 2 x Intel Xeon Platinum 826 (Cascade Lake) 2.9 GHz 24-core processors, 48 cores and 192 GB DDR4-2933 RAM per node. To illustrate our proposed algorithm and facilitate code adoption, we provide a step-by-step Jupyter notebook outlining RandNet-Parareal. Moreover, all simulation outcomes, including tables and figures, are fully reproducible and accompanied by the necessary Python code at https://github.com/Parallel-in-Time-Differential-Equations/RandNet-Parareal.

It is well acknowledged that comparing PinT methods based on different working principles is extremely hard, with [55] representing a recent survey article with some comparisons. Quoting [55],"caution should be taken when directly comparing speedup numbers across methods and implementations. In particular, some of the speedup and efficiency numbers are only theoretical in nature, and many of the parallel time methods do not address the storage or communication overhead of the parallel time integrator". [19] is one of very few recent attempts to systematically compare different PinT classes. However, it is limited exclusively to the Dahlquist problem. Thus, it has become conventional to compare new techniques to the existing state-of-the-art methods within the same group of solvers. This is why, in this work, we compare RandNet-Parareal with the original Parareal and its recently improved versions, GParareal [57], and nnGParareal [21].

The rest of the paper is organized as follows. In Section 2, we describe the Parareal algorithm. Section 3 briefly explains GParareal and nnGParareal, focusing on the latter. RandNet-Parareal is introduced in Section 4, while Sections 5 and 6 present our numerical results, and a final discussion. A computational complexity analysis of RandNet-Parareal, a robustness evaluation of the proposed algorithm, complementary simulation studies, and other additional results are available in the Supplementary Material.

**Notation.** We denote by $\boldsymbol{v} \in \mathbb{R}^n$ a column vector with entries $v_i$, $i \in \{1, \ldots, n\}$, and by $\|\boldsymbol{v}\|$ and $\|\boldsymbol{v}\|_\infty$ its Euclidean and infinity norms, respectively. We use $A \in \mathbb{R}^{n \times m}$ to denote a real-valued $n \times m$ matrix, $n, m \in \mathbb{N}$, with elements $A_{ij}$, $j$th column $A_{(\cdot, j)}$, $j \in \{1, \ldots m\}$, and $i$th row $A_{(i, \cdot)}$, $i \in \{1, \ldots, n\}$. We write $A^\top$, $A^\dagger$, and $\|A\|_{\mathrm{F}}$ for the $A$ matrix transpose, Moore-Penrose pseudoinverse, and Frobenius norm, respectively. $\mathbb{I}_n$ denotes the identity matrix of dimension $n$.

## 2 The Parareal algorithm

The idea of Parareal is to solve the $d$-dimensional ODE (and similarly PDE) system (1) in a parallel-in-time fashion, dividing the original IVP into $N$ sub-IVPs

$$\frac{d\boldsymbol{u}_i}{dt} = h\left(\boldsymbol{u}_i\left(t \mid \boldsymbol{U}_i\right), t\right), \quad t \in [t_i, t_{i+1}], \quad \boldsymbol{u}_i\left(t_i\right) = \boldsymbol{U}_i, \quad \text{for } i = 0, \ldots, N-1,$$

where the number of time intervals $N$ is also the number of available machines/cores/processors, $\boldsymbol{u}_i\left(t \mid \boldsymbol{U}_i\right)$ is the solution at time $t$ of the $i^{\text{th}}$ IVP with initial condition $\boldsymbol{u}(t_i) = \boldsymbol{U}_i \in \mathbb{R}^d$, $i = 0, \ldots, N-1$. If the initial conditions were known and satisfied the continuity conditions $\boldsymbol{U}_i = \boldsymbol{u}_{i-1}\left(t_i \mid \boldsymbol{U}_{i-1}\right)$ (for the coherent temporal evolution of the system across sub-intervals), then the sub-IVPs could be trivially solved in parallel on a dedicated machine. Unfortunately, this is not the case, as only the first initial condition $\boldsymbol{U}_0 = \boldsymbol{u}^0 \in \mathbb{R}^d$ at time $t_0$ appears available. To account for this, Parareal introduces another numerical integrator $\mathscr{G}$, much faster but less accurate than $\mathscr{F}$, to approximate the missing initial conditions $\boldsymbol{U}_i$, $i = 1, \ldots, N-1$, *sequentially*. $\mathscr{G}$ trades off accuracy for computational feasibility, usually taking seconds/minutes instead of hours/days of $\mathscr{F}$[2].

The algorithm works as follows. We use $\boldsymbol{U}_i^k$ to denote the Parareal approximation of $\boldsymbol{u}_i(t_i) = \boldsymbol{U}_i$ at iteration $k \geq 0$. At $k = 0$, the initial conditions $\{\boldsymbol{U}_i^0\}_{i=1}^{N-1}$ are initialized using a *sequential* application of the coarse solver $\mathscr{G}$, obtaining $\boldsymbol{U}_i^0 = \mathscr{G}(\boldsymbol{U}_{i-1}^0)$, $i = 1, \ldots, N-1$, with $\boldsymbol{U}_0^0 = \boldsymbol{U}_0$. At $k \geq 1$, the obtained initial conditions $\boldsymbol{U}_{i-1}^{k-1}$ are "propagated" through $\mathscr{F}$ in *parallel* on $N$ cores to obtain $\mathscr{F}(\boldsymbol{U}_{i-1}^{k-1})$, $i = 1, \ldots, N$. Note that for every initial condition $\boldsymbol{U}_{i-1}^{k-1}$, we compute both $\mathscr{F}(\boldsymbol{U}_{i-1}^{k-1})$, i.e. a precise evaluation of $\boldsymbol{u}_{i-1}(t_i \mid \boldsymbol{U}_{i-1}^{k-1})$, and $\mathscr{G}(\boldsymbol{U}_{i-1}^{k-1})$, an inaccurate evaluation of the same term. Hence, we can interpret $\mathscr{F}$ and $\mathscr{G}$ as functions mapping an initial condition to the next one, thereby evolving (1) by one interval. We can then use their difference, $(\mathscr{F} - \mathscr{G})(\boldsymbol{U}_{i-1}^{k-1})$, to correct the inaccuracy of $\mathscr{G}$ on future evaluations. This gives rise to the original Parareal predictor-corrector rule $\boldsymbol{U}_i^k = \mathscr{G}(\boldsymbol{U}_{i-1}^k) + (\mathscr{F} - \mathscr{G})(\boldsymbol{U}_{i-1}^{k-1})$, with $i = 1, \ldots, N-1$, $k \geq 1$ [18], where the *sequential* prediction $\mathscr{G}(\boldsymbol{U}_{i-1}^k)$ is corrected by adding the discrepancy $\mathscr{F} - \mathscr{G}$ computed at the previous iteration $k - 1$. However, this formulation can be changed to use data from the current iteration $k$ [57], and generalized to account for different ways of computing the discrepancy, leading to [21]

$$\boldsymbol{U}_i^k = \mathscr{G}(\boldsymbol{U}_{i-1}^k) + \widehat{f}(\boldsymbol{U}_{i-1}^k), \tag{2}$$

where $\widehat{f} : \mathbb{R}^d \to \mathbb{R}^d$ specifies how the correction function $\mathscr{F} - \mathscr{G}$ is computed or approximated based on some observation $\boldsymbol{U} \in \mathbb{R}^d$. Parareal uses

$$\widehat{f}_{\text{Para}}(\boldsymbol{U}_{i-1}^k) = (\mathscr{F} - \mathscr{G})(\boldsymbol{U}_{i-1}^{k-1}), \tag{3}$$

while other variants will be introduced in the subsequent sections. The Parareal solution (2) is considered converged for a given threshold $\epsilon > 0$ and up to time $t_L \leq t_N$, if solutions across consecutive iterations have stabilized. That is, for some pre-defined accuracy level $\epsilon > 0$, it holds that

$$\|\boldsymbol{U}_i^k - \boldsymbol{U}_i^{k-1}\|_\infty < \epsilon, \quad 0 < i \leq L \leq N - 1. \tag{4}$$

Other stopping criteria are also possible [66, 67]. Converged Parareal approximations $\boldsymbol{U}_i^k$, $i \leq L$, are no longer iterated to avoid unnecessary overhead [12, 20, 21, 57, 58]. Instead, unconverged solution values $\boldsymbol{U}_i^k$, $i > L$, are updated during future iterations by first running $\mathscr{F}$ in parallel and then using the prediction-correction rule (2). The Parareal algorithm stops at some iteration $K_{\text{Para}} \leq N$ when all initial conditions have converged, that is when (4) is satisfied with $L = N - 1$ and thus $K_{\text{Para}} = k$. Note that during every Parareal iteration $k > 1$, the "leftmost" fine solver evaluation $\mathscr{F}(\boldsymbol{U}_L^k)$ is either run from the outcome of a previous fine computation $\boldsymbol{U}_L^k = \mathscr{F}(\boldsymbol{U}_{L-1}^{k-1})$, or from a converged initial condition $\|\boldsymbol{U}_L^k - \boldsymbol{U}_L^{k-1}\|_\infty < \epsilon$. This guarantees that, either way, the maximum number of iterations to convergence for *any* Parareal-based algorithm is $K_{\text{Para}} = N$, in which case it sequentially attains the fine solver solution, with the added computational cost of running $\mathscr{G}$ and evaluating $\widehat{f}$ $N$ times. A Parareal pseudocode is presented in Algorithm 1 in Supplementary Material A.

## 3   GParareal and Nearest Neighbors GParareal

The performance of Parareal can be improved by a careful selection of $\widehat{f}$ in (2), combined with a better use of the available information present at iteration $k$. Let $\mathcal{D}_k$ denote the dataset consisting of $Nk$ pairs of inputs $\boldsymbol{U}_{i-1}^j \in \mathbb{R}^d$ and their corresponding outputs $(\mathscr{F} - \mathscr{G})(\boldsymbol{U}_{i-1}^j) \in \mathbb{R}^d$, $i = 1, \ldots, N$, $j = 0, \ldots, k-1$, that is

$$\mathcal{D}_k := \{(\boldsymbol{U}_{i-1}^j, (\mathscr{F} - \mathscr{G})(\boldsymbol{U}_{i-1}^j)), \ i = 1, \ldots, N, \ j = 0, \ldots, k-1\}. \tag{5}$$

---

[2]$\mathscr{F}$ and $\mathscr{G}$ can be two different solvers or the same solver with different time steps.

While Parareal relies on one observation to construct the correction $\widehat{f}$ in (3), GParareal and following works, including this one, use all the discrepancy terms $\mathscr{F} - \mathscr{G}$ and information in $\mathcal{D}_k$ to make their predictions. The idea of GParareal is to learn the map $\mathbb{R}^d \to \mathbb{R}^d$, $\boldsymbol{U}_{i-1}^k \mapsto (\mathscr{F} - \mathscr{G})(\boldsymbol{U}_{i-1}^k)$, via $d$ independent scalar GPs $\mathbb{R}^d \to \mathbb{R}$, $\boldsymbol{U}_{i-1}^k \mapsto \widehat{f}_{\text{GPara}}^{(s)}(\boldsymbol{U}_{i-1}^k)$, $s = 1, \ldots, d$, one per ODE dimension, whose predictions are concatenated into $\widehat{f}_{\text{GPara}}(\boldsymbol{U}_{i-1}^k) = (\widehat{f}_{\text{GPara}}^{(1)}(\boldsymbol{U}_{i-1}^k), \ldots, \widehat{f}_{\text{GPara}}^{(d)}(\boldsymbol{U}_{i-1}^k))^\top \in \mathbb{R}^d$, and finally plugged into the predictor-corrector rule (2). In particular, each GP prediction $\widehat{f}_{\text{GPara}}^{(s)}(\boldsymbol{U}_{i-1}^k)$ is obtained as the GP posterior mean $\mu_{\mathcal{D}_k}^{(s)}(\boldsymbol{U}_{i-1}^k) \in \mathbb{R}$, computed by conditioning the corresponding GP prior on the dataset $\mathcal{D}_k$, i.e. $\widehat{f}_{\text{GPara}}^{(s)}(\boldsymbol{U}_{i-1}^k) = \mu_{\mathcal{D}_k}^{(s)}(\boldsymbol{U}_{i-1}^k)$. We refer to Supplementary Material B and [57] for a thorough description of the algorithm, including all relevant quantities of interest, namely the $d$ GP priors, the likelihood, the hyperparameters and their optimization procedure, and an explicit expression of the posterior means. Here, it is worth highlighting that the GPs are trained once per iteration to leverage the new incoming data, and then their predictions are used to *sequentially* update the initial conditions in (2). Using all information stored in $\mathcal{D}_k$ instead of a single observation (as for Parareal) is the primary driver of faster convergence rates experienced by GParareal. Other benefits of this algorithm are increased stability to different initial conditions, the ability to incorporate legacy data (that is, the possibility of using datasets coming from previous runs of the algorithm with different starting conditions or settings, leading to faster convergence), lower sensitivity to poor choices of the coarse solver $\mathscr{G}$, and the possibility of parallelizing the training of the $d$ GPs over the $N$ available cores. The main drawback of GParareal is the heavy computational burden incurred when inverting the GP covariance matrices, which is of order $O(d(Nk)^3)$ at iteration $k$. This negatively impacts the algorithm's wallclock time, which may be higher than Parareal despite a lower number of iterations needed to converge. This is why GParareal has been proposed mainly for low-dimensional ODE systems with a relatively small number of processors/intervals $N$ (up to hundreds), limiting its use and parallel scalability [57].

The nnGParareal algorithm [21] has been proposed to tackle GParareal's scalability issue, sensibly reducing the computational time and memory footprint of GPs by using their nns version (nnGPs). In this framework, at iteration $k$, the $d$ GPs are all trained on a smaller dataset of size $m$, $\mathcal{D}_{i-1,k}$, composed out of the $m$ nns (in Euclidean distance) of $\boldsymbol{U}_{i-1}^k$ in $\mathcal{D}_k$, leading to the nnGParareal correction $\widehat{f}_{\text{nnGPara}}(\boldsymbol{U}_{i-1}^k) = (\widehat{f}_{\text{nnGPara}}^{(1)}(\boldsymbol{U}_{i-1}^k), \ldots, \widehat{f}_{\text{nnGPara}}^{(d)}(\boldsymbol{U}_{i-1}^k))^\top$, with

$$\widehat{f}_{\text{nnGPara}}^{(s)}(\boldsymbol{U}_{i-1}^k) = \mu_{\mathcal{D}_{i-1,k}}^{(s)}(\boldsymbol{U}_{i-1}^k), \quad s = 1, \ldots, d.$$

Here, $\mu_{\mathcal{D}_{i-1,k}}^{(s)} \in \mathbb{R}$, $s = 1, \ldots, d$, denotes the nnGP posterior mean computed by conditioning the corresponding GP prior on the reduced dataset $D_{i-1,k}$ of size $m$. Due to the decreased sample size, each nnGP covariance matrix can be inverted at a cost of $O(m^3)$ independent of $k$ or $N$. However, contrary to GParareal which trains the GPs once per iteration, the nnGPs are re-trained *every time a new prediction* $\widehat{f}_{\text{nnGPara}}(\boldsymbol{U}_{i-1}^k)$ *is made*, which are at most $N - k$ at iteration $k$ (as at least $k$ intervals have converged at iteration $k$), yielding a combined $O(d(N - k)m^3)$ complexity. Several experiments on different ODE and PDE systems have shown that $m \in \{15, \ldots, 20\}$ offer accuracy comparable to the full GP [21] at a much lower cost. Although faster than GParareal, nnGParareal still exhibits some of the drawbacks inherited from the GP framework, such as the cost of optimizing the hyperparameters through a numerical maximization of a non-convex likelihood, and the use of $d$ scalar nnGPs. The latter is particularly critical. On the one hand, despite the possibility of training the $d$ nnGPs in parallel, the inversion of a $m \times m$ matrix is so efficient that parallel overheads may outweigh the theoretical benefits. On the other hand, when solving PDEs, nnGParareal will incur additional costs due to insufficient hardware resources, as usually $d \gg N$, forcing the $d$ nnGPs to queue among the $N$ available processors, which is why the algorithm has been proposed for high-dimensional ODE and PDE systems with $d \leq N$. We refer to Supplementary Material B and [21] for more details on nnGParareal, and to Algorithm 2 in Supplementary Material A for the pseudocode of the nnGP training. In the next section, we address the nnGParareal issues by introducing RandNets.

## 4  Random neural networks Parareal (RandNets-Parareal)

In RandNet-Parareal, we propose to learn the map $\mathbb{R}^d \to \mathbb{R}^d$, $\boldsymbol{U} \mapsto (\mathscr{F} - \mathscr{G})(\boldsymbol{U})$ via RandNets, obtaining the RandNet-Parareal correction $\widehat{f}_{\text{RandNet-Para}}$, which we then use within the predictor-corrector rule (2). Prior to that, we define how RandNets work in a general setting with input $\boldsymbol{U} \in \mathbb{R}^d$

and output or target $\boldsymbol{Y} \in \mathbb{R}^d$. Later in the text we will go back to the input of interest $\boldsymbol{U}_i^k$. Let $M$ denote the number of hidden neurons, and $H_W^{A;\boldsymbol{\zeta}}(\boldsymbol{U})$ be a single-hidden-layer feed-forward neural network used to learn $\mathscr{F} - \mathscr{G}$, given by

$$H_W^{A;\boldsymbol{\zeta}}(\boldsymbol{U}) = W^\top \boldsymbol{\sigma}(A\boldsymbol{U} + \boldsymbol{\zeta}) \in \mathbb{R}^d, \quad \boldsymbol{U} \in \mathbb{R}^d, \tag{6}$$

where $A \in \mathbb{R}^{M \times d}$ is the matrix of random, non-trainable weights of the hidden layer, $\boldsymbol{\zeta} \in \mathbb{R}^M$ is a random non-trainable bias vector, and $W \in \mathbb{R}^{M \times d}$ is the matrix of trainable output weights. Here, $\boldsymbol{\sigma} : \mathbb{R}^M \to \mathbb{R}^M$ denotes an activation function obtained as the componentwise application of a non-linear map $\sigma : \mathbb{R} \to \mathbb{R}$ which we choose to be ReLU $\sigma(x) = \max(x, 0)$ with $x \in \mathbb{R}$, to satisfy the assumption of Proposition 1 below. The entries of $A$ and $\boldsymbol{\zeta}$ are randomly sampled from given distributions $\mathcal{P}_A$ and $\mathcal{P}_{\boldsymbol{\zeta}}$, respectively, and kept fixed. After observing the dataset $\mathcal{D}_k$, the output weights $W$ are obtained as the minimum $\ell_2$ norm least squares (or simply min-norm least squares) estimator or as the solution of the following penalized empirical minimization problem:

$$\widehat{W}^{\mathcal{D}_k} = \lim_{\lambda \to 0} \arg \min_{W \in \mathbb{R}^{M \times d}} \left\{ \sum_{(\boldsymbol{U}, \boldsymbol{Y}) \in \mathcal{D}_k} \left\| H_W^{A;\boldsymbol{\zeta}}(\boldsymbol{U}) - \boldsymbol{Y} \right\|^2 + \lambda \|W\|_{\mathrm{F}}^2 \right\},$$

which is also called a "ridgeless" (interpolation) estimator [30], and can be more compactly written as

$$\widehat{W}^{\mathcal{D}_k} = \lim_{\lambda \to 0} \left( X^\top X + \lambda \mathbb{I}_M \right)^{-1} X^\top Y. \tag{7}$$

Here, $X \in \mathbb{R}^{Nk \times M}$ is a matrix with $(X_{(l,\cdot)})^\top := \boldsymbol{\sigma}(A(U_{(l,\cdot)})^\top + \boldsymbol{\zeta})$, $l = 1, \ldots, Nk$, and $U, Y \in \mathbb{R}^{Nk \times d}$ are the collection of inputs and outputs of $\mathcal{D}_k$ in matrix form, respectively, defined as $(U_{(l,\cdot)})^\top = \boldsymbol{U}_i^j$, $(Y_{(l,\cdot)})^\top = \boldsymbol{Y}_i^j$, $l = jN + i + 1$, $i = 0, \ldots, N-1$, $j = 0, \ldots, k-1$. Whenever $Nk \geq M$ and the rank of $X^\top X \in \mathbb{R}^{M \times M}$ is $M$, (7) reduces to the standard least squares estimator $\widehat{W}^{\mathcal{D}_k} = \left( X^\top X \right)^{-1} X^\top Y$, while if the rank of $X^\top X$ is $Nk$, the solution admits a closed form

$$\widehat{W}^{\mathcal{D}_k} = \left( X^\top X \right)^\dagger X^\top Y.$$

We get inspired by [21], where only $m$ nns are used in the training. In this setting, $M \gg Nk = m$, and in this overparametrized linear regression case, the ridgeless estimator interpolates the training data, which is a desirable feature since the problem is genuinely deterministic [29, 49].

Several ingredients control the performance of RandNets, such as the dimension of the network $M$ and the choice of distributions $\mathcal{P}_A$ and $\mathcal{P}_{\boldsymbol{\zeta}}$. In this work, we take the rows of the weight matrix $A$ and the bias entries of $\boldsymbol{\zeta}$ to be independent and uniformly distributed. For this case, the approximation bounds are available [25, Proposition 3], which we report below using our notation.

**Proposition 1** (Approximation bound, [25], Proposition 3). *Let $H^* : \mathbb{R}^d \to \mathbb{R}$, $\boldsymbol{U} \longmapsto H^*(\boldsymbol{U})$ be an unknown function we wish to approximate with $H_W^{A;\boldsymbol{\zeta}}$ defined in (6). Suppose $H^*$ can be represented as $H^*(\boldsymbol{U}) = \int_{\mathbb{R}^d} e^{i\langle \mathbf{w}, \boldsymbol{U} \rangle} g(\mathbf{w}) \mathrm{d}\mathbf{w}$ for some complex-valued function $g$ on $\mathbb{R}^d$ and all $\boldsymbol{U} \in \mathbb{R}^d$ with $\|\boldsymbol{U}\| \leq Q$, where $\langle \cdot, \cdot \rangle$ is the inner product on $\mathbb{R}^d$. Assume that $\int_{\mathbb{R}^d} \max\left(1, \|\mathbf{w}\|^{2d+6}\right) |g(\mathbf{w})|^2 \, \mathrm{d}\mathbf{w} < \infty$. For $\rho > 0$, suppose the rows of $A$ are i.i.d. random variables with uniform distribution on $B_\rho \subset \mathbb{R}^d$, the Euclidean ball of radius $\rho$ around $\mathbf{0}$, and that the $M$ components of $\boldsymbol{\zeta}$ are i.i.d. uniform random variables on $[-\max(Q\rho, 1), \max(Q\rho, 1)]$. Assume that $A$ and $\boldsymbol{\zeta}$ are independent and let $\sigma : \mathbb{R} \to \mathbb{R}$ be given by $\sigma(x) = \max(x, 0)$. Then, there exist a $\mathbb{R}^{M \times d}$-valued random variable $W$ and an explicit (see (33) in [25]) constant $C^* > 0$ such that*

$$\mathbb{E}\left[ \|H_W^{A;\boldsymbol{\zeta}}(\boldsymbol{U}) - H^*(\boldsymbol{U})\|^2 \right] \leq \frac{C^*}{M},$$

*and for any $\delta \in (0, 1)$, the random neural network $H_W^{A;\boldsymbol{\zeta}}$ satisfies*

$$\mathbb{P}\left( \left( \int_{\mathbb{R}^d} \|H_W^{A;\boldsymbol{\zeta}}(\boldsymbol{U}) - H^*(\boldsymbol{U})\|^2 \mu_{\boldsymbol{U}}(\mathrm{d}\boldsymbol{U}) \right)^{1/2} \leq \frac{\sqrt{C^*}}{\delta\sqrt{M}} \right) \geq 1 - \delta.$$

Our choice of $\mathcal{P}_A$ and $\mathcal{P}_{\boldsymbol{\zeta}}$ satisfies the conditions of Proposition 1 if $\|\boldsymbol{U}\| \leq Q$. If this is not met, we rescale the ODE/PDE system via a change of variables. We found these bounds empirically useful in informing a good choice for the sampling distribution, which we follow. If no prior

information were available, the common approach would have been to take $\mathcal{P}_A \sim \text{Unif}(-a, a)^{M \times d}$, $\mathcal{P}_{\boldsymbol{\zeta}} \sim \text{Unif}(-b, b)^M$, and optimize $a, b \in \mathbb{R}^+$ via expensive cross-validation procedure.

Unlike nnGParareal, GParareal, and the corresponding nnGPs and GPs, training RandNets is so fast that parallelization across the $d$ dimensions is unnecessary. Hence, the predictions of the random network are computed jointly on all $d$ coordinates, yielding the RandNet-Parareal correction function

$$\widehat{f}_{\text{RandNet-Para}}(\boldsymbol{U}_{i-1}^k) = H_{\widehat{W}^{\mathcal{D}_{i-1,k}}}^{A, \boldsymbol{\zeta}}(\boldsymbol{U}_{i-1}^k) \in \mathbb{R}^d. \tag{8}$$

Here, the estimated weights $\widehat{W}^{\mathcal{D}_{i-1,k}}$ are obtained using the reduced dataset $\mathcal{D}_{i-1,k}$ consisting of the $m_{\text{RandNet}}$ nns of $\boldsymbol{U}_{i-1}^k$, requiring the retraining of the RandNet for every prediction. Employing a multi-output model instead of independently training $d$ scalar-output models addresses one of the pitfalls of GPs, allowing for better scalability when $d \gg N$. The fact that training the RandNets reduces to a closed-form ridgeless interpolation solution presents a substantial difference and improvement with respect to (nn)GPs. Moreover, expensive hyperparameter optimization is avoided in RandNets, addressing the other major pitfall of GParareal and nnGParareal. The pseudocode for training RandNets is reported in Algorithm 3 in Supplementary Material A.

In Supplementary Material C, we derive the theoretical computational costs of nnGParareal and RandNet-Parareal, illustrating them as a function of dimension $d$ and number of processors $N$ in Figure 3. These theoretical findings confirm the significantly superior scalability of RandNet-Parareal which we observe in the numerical experiments reported in Section 5.

In Supplementary Material D, we study the robustness of RandNet-Parareal to changes in the number of nns $m_{\text{RandNet}}$ (and thus the input data size), the number of neurons $M$, and the randomly sampled network weights $A, \boldsymbol{\zeta}$. Intuitively, one might anticipate that a larger data sample would yield a more accurate approximation of the correction $\mathscr{F} - \mathscr{G}$, and that a higher number of neurons $M$ would reduce the prediction error of RandNets (as in Proposition 1). One may also suspect the algorithm to be sensitive to the particular sampling seed. Remarkably, our empirical findings demonstrate that these factors have a limited impact on the number of iterations needed by RandNet-Parareal to converge, which remains largely consistent (up to a few iterations) across different values and ODE/PDE systems, for sensible choices of $m_{\text{RandNet}}$ and $M$. For the end user, this eliminates the need of ad-hoc tuning, making the proposed RandNet-Parareal a convenient out-of-the-box algorithm.

## 5 Numerical Experiments

In this section, we first compare the performance of Parareal, nnGParareal, and RandNet-Parareal on the viscous Burgers' equation (one spatial dimension and one variable, also considered in nnGParareal [21]), to showcase Parareal and nnGParareal challenges as the number of space discretization and, correspondingly, the dimensions $d$, increases. Then, we consider the Diffusion-Reaction equation, a larger system defined on a two-dimensional spatial domain with two non-linearly coupled variables, and the SWEs (two spatial dimensions and three variables), representing a suitable framework for modeling free-surface flow problems on a two-dimensional domain. Two additional challenging systems, the 2D and 3D Brusselator PDEs, known for their complex behavior, including oscillations, spatial patterns, and chaos, are considered in Supplementary Material E. The simulation setups used for obtaining the results in this section are provided in Supplementary Material G, with the corresponding accuracies and runtimes for RandNet-Parareal, Parareal, and nnGParareal reported in Supplementary Material F.

Let $T_{\mathscr{F}}$ and $T_{\mathscr{G}}$ be the time it takes to run $\mathscr{F}$ and $\mathscr{G}$ over one interval $[t_i, t_{i+1}]$, respectively, and let $N_{\mathscr{F}}$ and $N_{\mathscr{G}}$ denote the number of steps for the fine and coarse solvers over one interval, respectively. We can measure the parallel efficiency of an algorithm via its parallel speed-up $S_{\text{alg}}$, defined as the ratio of the serial over the parallel runtime, i.e. $S_{\text{alg}} := N T_{\mathscr{F}} / T_{\text{alg}}$. $S_{\text{alg}}$ captures the wallclock gains of parallel procedures and, unlike other quantities (such as the number of algorithm iterations needed to converge), also includes the model training cost.

### 5.1 Viscous Burgers' equation

Our initial example is a non-linear, one-dimensional PDE (illustrated in Figure 7 of Supplementary Material H) exhibiting hyperbolic behavior [68], described by the equation

$$v_t = \nu v_{xx} - v v_x, \quad (x, t) \in [-L, L] \times [t_0, t_N], \tag{9}$$

Table 1: Empirical scalability and speed-up analysis for viscous Burgers' equation

| $d = 128, N = 128$ | | | | | | |
| --- | --- | --- | --- | --- | --- | --- |
| Algorithm | $K$ | $NT_{\mathcal{G}}$ | $T_{\mathcal{F}}$ | $T_{\text{model}}$ | $T_{\text{alg}}$ | $S_{\text{alg}}$ |
| Fine | – | – | – | – | 13h 5m | 1 |
| Parareal | 90 | 0s | 6m | 0s | 8h 54m | 1.47 |
| nnGParareal | 14 | 0s | 6m | 12m | 1h 39m | 7.90 |
| RandNet-Parareal | 10 | 0s | 6m | 1s | 1h 2m | **12.61** |

| $d = 1128, N = 128$ | | | | | | |
| --- | --- | --- | --- | --- | --- | --- |
| Algorithm | $K$ | $NT_{\mathcal{G}}$ | $T_{\mathcal{F}}$ | $T_{\text{model}}$ | $T_{\text{alg}}$ | $S_{\text{alg}}$ |
| Fine | – | – | – | – | 18h 52m | 1 |
| Parareal | 91 | 0s | 9m | 0s | 12h 57m | 1.41 |
| nnGParareal | 6 | 2s | 9m | 1h 25m | 2h 17m | 8.26 |
| RandNet-Parareal | 4 | 2s | 9m | 1s | 38m | **29.98** |

Speed-up $S_{\text{alg}}$ of Parareal, nnGParareal ($m_{\text{nnGP}}$=18), and RandNet-Parareal ($m_{\text{RandNet}}$=4, $M$=100) for the 1D viscous Burgers' equation. $T_{\mathcal{F}}$ and $T_{\mathcal{G}}$ are the interval runtimes of the fine and coarse solvers, respectively, $K$ the number of iterations to converge, $T_{\text{model}}$ the overall time to evaluate $\widehat{f}$ across $K$ iterations, including training and predicting, and $T_{\text{alg}}$ thealgorithm runtime.

with initial condition $v(x, t_0) = v_0(x)$, $x \in [-L, L], L > 0$, and Dirichlet boundary conditions $v(-L, t) = v(L, t)$, $v_x(-L, t) = v_x(L, t)$, $t \in [t_0, t_N]$. We use the same setting and parameter values as in [21]. More specifically, we choose $L = 1$, diffusion coefficient $\nu = 0.01$, and discretize the spatial domain using finite difference [15] and equally spaced points $x_{j+1} = x_j + \Delta x$, with $\Delta x = 2L/d$ and $j = 0, \ldots, d$. We hence reformulate the PDE as a $d$-dimensional ODE system.

In our first numerical experiment, we choose $N = d = 128$, $v_0(x) = 0.5(\cos(\frac{9}{2}\pi x) + 1)$, $t_0 = 0$, and $t_N = 5.9$ as in [21], and consider $\mathcal{G} = \text{RK1}$, $\mathcal{F} = \text{RK8}$, $N_{\mathcal{G}} = 4$ and $N_{\mathcal{F}} = 4e^4$, where RK1 stands for Runge-Kutta of order 1, and similarly for RK4 and RK8. The results, reported at the top of Table 1, show how RandNet-Parareal converges in fewer iterations and has a higher speed-up than Parareal and nnGParareal. The difference in the model training costs is striking, with the nnGP's being approximately 700 times higher than that of RandNets, reducing thus its potential speed-up.

As real-world (one-dimensional) problems would require a higher spatial discretization, we increase $d$ by one thousand to $d = 1128$, keeping $N$ fixed. Unlike assuming matching hardware resources to the system size (as implicitly done in [21], where $d = N$), we deliberately do not increase $N$ to assess the algorithms' performances under constrained conditions. Instead, both time discretization numbers are increased to $N_{\mathcal{F}} = 6e^5$ and $N_{\mathcal{G}} = 293$ (resulting thus in longer $T_{\mathcal{F}}$ and $T_{\mathcal{G}}$ times) to account for the finer spatial mesh [43]. As observed from the bottom of Table 1, as $d/N > 1$, nnGParareal's issues become more pronounced, as the $d$ scalar GPs cannot be run all in parallel across the $N$ processors, but need $d/N = 10$ runs instead, slowing down the algorithm. In contrast, RandNet-Parareal has a training cost comparable with the previous example, leading to an even higher speed-up, running in approximately 38 minutes compared to the almost 13 hours of Parareal.

### 5.2 Diffusion-Reaction system

We now turn to a more challenging case study. The Diffusion-Reaction equation [75] (illustrated in Figure 8 in Supplementary Material H) is a system of two non-linearly coupled variables, the activator $u = u(t, x, y)$ and the inhibitor $v = v(t, x, y)$, defined on a two-dimensional spatial domain as

$$\partial_t u = D_u \partial_{xx} u + D_u \partial_{yy} u + R_u, \quad \partial_t v = D_v \partial_{xx} v + D_v \partial_{yy} v + R_v.$$

Here, $D_u, D_v$ are the diffusion coefficients for the activator and inhibitor, respectively, and $R_u = R_u(u, v)$, $R_v = R_v(u, v)$ are their reaction functions defined by the Fitzhugh-Nagumo equation [42]

$$R_u(u, v) = u - u^3 - c - v, \qquad R_v(u, v) = u - v,$$

where $c = 5e^{-3}$, $D_u = 1e^{-3}$, and $D_v = 5e^{-3}$. We take $(x, y) \in (-1, 1)^2$ and $t \in [0, 20]$. The initial condition $u(0, x, y)$ is generated as standard Gaussian noise. We apply a no-flow Neumann boundary

condition $D_u \partial_x u = 0$, $D_v \partial_x v = 0$, $D_u \partial_y u = 0$, $D_v \partial_y v = 0$ for $(x, y) \in (-1, 1)^2$. The spatial domain is discretized by the finite volume method [51], resulting in a $d = 2N_x N_y$-dimensional ODE with $N_x$ and $N_y$ the number of space discretizations along $x$ and $y$, respectively. The time integration is conducted with RK of variable order for $\mathscr{G}$ and $\mathscr{F}$ (see Table 6 in Supplementary Material G).

As in the previous example, we conduct two experiments for this system, with speed-ups and runtimes reported in Figure 1. In the first one, we increased $d$ and $N$ proportionately (with $d/N \in [11, 13]$) while maintaining all other quantities (i.e. $\mathscr{G}, \mathscr{F}, m_{\mathrm{nnGP}}, m_{\mathrm{RandNet}}$) fixed until $N = 256$. This scenario reflects a situation where more resources are allocated to solve larger problem sizes. In contrast, in the second experiment, $N$ remains fixed at $512$, with $d$ increasing proportionately with $N_{\mathscr{G}}$ to maintain algorithm stability. Moreover, $\mathscr{F}$ is chosen to be RK8, with $N_{\mathscr{F}}$ automatically selected by the used Python library *scipy* [77]. This second setting simulates a scenario with constrained resources, where the user aims to solve the system using a finer spatial mesh. Table 8 in Supplementary Material I shows that for $N \geq 256$ and $d/N \gg 1$, nnGParareal fails to converge within a 48-hour budget. Parareal converges always, albeit at a considerably slower rate than RandNet-Parareal, which is x3-5 faster than Parareal (and up to x120 than the fine solver).

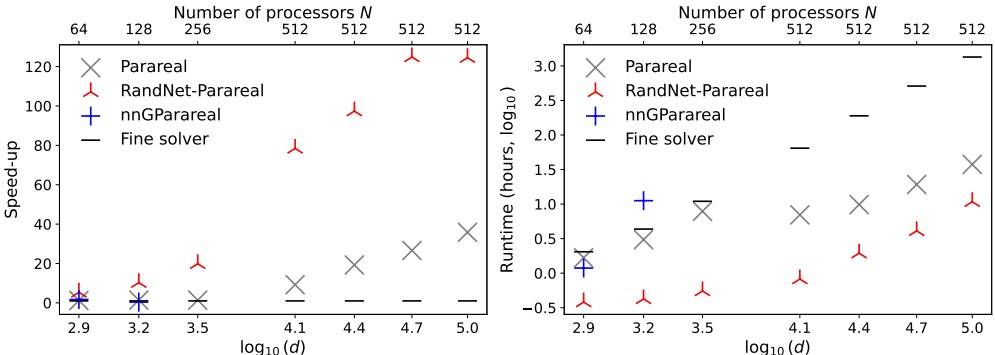

Figure 1: Speed-ups (left) and runtimes (right) of Parareal, nnGParareal ($m_{\mathrm{nnGP}}$=20), and RandNet-Parareal ($m_{\mathrm{RandNet}}$=4, $M$=100) for the two-dimensional Diffusion-Reaction system versus the number $d$ of dimensions (bottom x-axis) and $N$ cores (top x-axis) capped at $512$ to simulate limited resources.

## 5.3   Shallow water equation

Finally, we focus on SWEs on a two-dimensional domain, described by a system of hyperbolic PDEs

$$\partial_t h + \nabla h \mathbf{u} = 0, \quad \partial_t h \mathbf{u} + \nabla(u^2 h + \tfrac{1}{2} g_r h^2) = -g_r h \nabla b,$$

where $\mathbf{u} = (u, v)$ represents the velocities in the horizontal $u = u(t, x, y)$ and vertical $v = v(t, x, y)$ directions, $h = h(t, x, y)$ denotes the water depth, $b = b(x, y)$ describes a (given) spatially varying bathymetry, and $h\mathbf{u}$ can be interpreted as the directional momentum components. The parameter $g_r$ describes the gravitational acceleration, while $\partial_t f$ denotes the partial derivative with respect to time, and $\nabla f$ the gradient of a function $f$. Following [75], we solve a radial dam break scenario where a Gaussian-shaped water column (blue) inundates nearby plains (green) within a rectangular box subject to Neumann boundary conditions, causing the water to rebound off the sides of the box, as depicted in Figure 2. More details on the simulation setup are given in Supplementary Material G.1.

In this case, our algorithm also converges much faster than Parareal, with a speed gain of x1.3-3.6, while nnGParareal fails to converge within the 48-hour time budget as $d \gg N$. Although the speed gain is lower than for the Diffusion-Reaction, the improvements are remarkable. RandNet-Parareal takes up to 4-10 hours and 37 days less than the Parareal and sequential solver, respectively.

## 6   Discussion and limitations

This study improves the scalability properties, convergence rates, and parallel performance of Parareal and a more recently proposed PinT solver for ODEs and PDEs, nnGParareal [21]. By replacing the

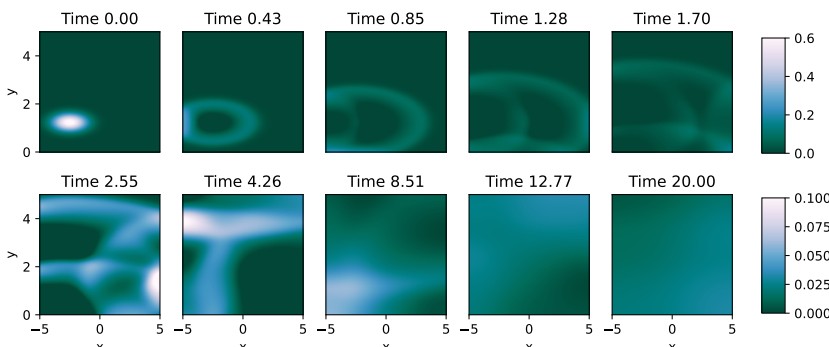

Figure 2: Numerical solution of the SWE for $(x, y) \in [-5, 5] \times [0, 5]$ with $N_x = 264$ and $N_y = 133$ for a range of system times $t$. Only the water depth $h$ (blue) is plotted.

Table 2: Speed-up analysis for the shallow water PDE as a $d$-dimensional ODE system, $N = 235$

| $d$ | $K_{\text{Para}}$ | $K_{\text{RandNet-Para}}$ | $T_{\mathscr{F}}$ | $T_{\text{Para}}$ | $T_{\text{RandNet-Para}}$ | $S_{\text{Para}}$ | $S_{\text{RandNet-Para}}$ |
|---|---|---|---|---|---|---|---|
| 15453 | 52 | 14 | 22h 54m | 5h 8m | 1h 25m | 4.47 | **16.16** |
| 31104 | 50 | 13 | 3d 2h | 15h 43m | 4h 9m | 4.68 | **17.69** |
| 60903 | 14 | 9 | 13d 15h | 19h 30m | 12h 34m | 16.73 | **25.92** |
| 105336 | 8 | 6 | 38d 4h | 1d 7h | 23h 34m | 29.37 | **38.90** |

$K.$ is the number of iterations to converge, $T.$ wallclock time and $S.$ speed-up for the Parareal (Para) and RandNet-Parareal ($m_{\text{RandNet}}$=4, $M$=100). $T_{\mathscr{F}}$ is the sequential runtime of $\mathscr{F}$. The results for nnGParareal ($m_{\text{nnGP}}$=20) are not reported as it fails to converge within a 48-hour time budget.

nnGP with random networks, we decreased the model costs (in learning the discrepancy between the fine and coarse solvers) by several orders of magnitude. The reasons behind this are multi-fold. Training of RandNets is cheap due to the availability of the closed-form solution for its output (readout) weights, and avoids any expensive hyperparameter optimization. Moreover, it is possible to simultaneously learn and predict the $d$-dimensional correction map instead of $d$ scalar maps (in parallel if the number of processors $N$ is comparable to $d$, or queuing if smaller). The latter "liberates" RandNet-Parareal from requiring $d \approx N$, extending its application to high-dimensional settings, a key/notable improvement with respect to nnGParareal. We tested the proposed algorithm on systems of real-world significance, such as the Diffusion-Reaction equation, the SWE, and the Brusselator. solving them on a fine spatial mesh of up to $10^5$ discretization points. These systems and requirements align with those outlined in the benchmark PDE dataset [75] as necessary prerequisites for using such algorithms in practical scenarios. The strength of RandNet-Parareal is the cheap cost of RandNets, which can be embedded within Parareal with virtually no overhead, irrespective of the implementation or solvers, leading to notable speed gains over Parareal (x8.6-21.2 for viscous Burgers', x3-5 for Diffusion-Reaction, x1.3-3.6 for SWE, and x3.4-4.4 for Brusselator). Moreover, training RandNets is easily conducted with established linear algebra routines, and requires no ad-hoc parameter tuning.

Despite its excellent performance, RandNet-Parareal has limitations common to all Parareal algorithms, as its rate of convergence relies on the accuracy of the coarse solver $\mathscr{G}$. Although neural networks can help mitigate the impact of suboptimal choices of $\mathscr{G}$ (as observed for GPs in (nn)GParareal), if the solver is mismatched for the system — for example, an unstable solver for a stiff ODE — RandNet-Parareal, similar to Parareal and (nn)GParareal, is likely to exhibit non-convergent behavior. It would then be of interest to investigate RandNet-Parareal's performance when using customized solvers tailored to specific systems, such as those outlined in Section 1 for the shallow water equation and the viscous Burgers' equation, which we defer to future research.

## Acknowledgments and Disclosure of Funding

GG is funded by the Warwick Centre of Doctoral Training in Mathematics and Statistics. GG thanks the hospitality of the University of St. Gallen where part of the results in this paper were obtained.

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

# A  Pseudocodes

This section provides pseudocodes for the implementation of Parareal (Algorithm 1), and the training procedure for learning the discrepancy $\mathscr{F} - \mathscr{G}$ via nnGPs in nnGParareal (Algorithm 2), and RandNets in RandNet-Parareal (Algorithm 3).

---

**Algorithm 1:** Parareal (generic)

---

**Input:** Initial condition $\boldsymbol{u}^0$ at time $t_0$, number of intervals $N$
**Output:** Converged initial conditions $\{\boldsymbol{U}_i^K\}_{i=1}^{N-1}$, with $K$ the number of iterations to
$\qquad$ convergence

**Initialization**
Rescale the ODE/PDE system such that each coordinate takes values in $[-1, 1]$
$L \leftarrow 1$
$\boldsymbol{U}_0^0 = \boldsymbol{u}^0$
**for** $i \leftarrow 1$ *to* $N - 1$ **do**
$\quad \mid \quad \boldsymbol{U}_i^0 \leftarrow \mathscr{G}(\boldsymbol{U}_{i-1}^0)$
**end**

**for** $k \leftarrow 1$ *to* $N$ **do**
$\quad \mid \quad$ Compute $\mathscr{F}(\boldsymbol{U}_{i-1}^{k-1})$, $i = 1, \ldots, N$ in *parallel*
$\quad \mid \quad$ **for** $i \leftarrow L + 1$ *to* $N - 1$ **do**
$\quad \mid \quad \mid \quad \boldsymbol{U}_i^k \leftarrow \mathscr{G}(\boldsymbol{U}_{i-1}^k) + \widehat{f}(\boldsymbol{U}_{i-1}^k)$ $\qquad$ /* Update the initial conditions */
$\quad \mid \quad$ **end**

$\quad \mid \quad$ **Convergence checks**
$\quad \mid \quad$ **for** $i \leftarrow L + 1$ *to* $N - 1$ **do**
$\quad \mid \quad \mid \quad$ **if** $\|\boldsymbol{U}_i^k - \boldsymbol{U}_i^{k-1}\|_\infty < \epsilon$ **then**
$\quad \mid \quad \mid \quad \mid \quad L \leftarrow L + 1$ $\qquad\qquad\qquad$ /* Update converged interval counter */
$\quad \mid \quad \mid \quad$ **else**
$\quad \mid \quad \mid \quad \mid \quad$ **break**
$\quad \mid \quad \mid \quad$ **end**
$\quad \mid \quad$ **end**

$\quad \mid \quad$ **if** $L == N$ **then**
$\quad \mid \quad \mid \quad$ **break** $\qquad\qquad\qquad\qquad$ /* All intervals have converged */
$\quad \mid \quad$ **end**
**end**

---

---

**Algorithm 2:** nnGP training procedure within nnGParareal

---

**Input:** Input $\boldsymbol{U}_{i-1}^k$, dataset $\mathcal{D}_k$, number of nearest neighbors $m_{\mathrm{nnGP}}$, number of random restarts for loss maximization $n_{\mathrm{start}}$
**Output:** Prediction $\widehat{f}_{\mathrm{nn}}(\boldsymbol{U}_{i-1}^k)$ of $(\mathscr{F} - \mathscr{G})(\boldsymbol{U}_{i-1}^k)$

**Initialization**
/* Find the $m_{\mathrm{nnGP}}$ nns to $\boldsymbol{U}_{i-1}^k$, and compute the reduced dataset (10) */
$\mathcal{D}_{i-1,k} \leftarrow \{(\boldsymbol{U}_{\boldsymbol{U}_{i-1}^k}^{(l\text{-nn})}, \mathbf{Y}_{\boldsymbol{U}_{i-1}^k}^{(l\text{-nn})}),\ l = 1, \dots, m_{\mathrm{nnGP}}\} \subset \mathcal{D}_k$

/* Both loops can be massively parallelized */
**for** $s \leftarrow 1$ **to** $N$ **do**
    /* Training */
    **for** $j \leftarrow 1$ **to** $n_{\mathrm{start}}$ **do**
        /* Random restarts to avoid local minima when maximizing (12) */
        Sample $\boldsymbol{\theta}_j^0$ at random
        Maximize (12) numerically using $\boldsymbol{\theta}_j^0$ as initial value; obtain $\boldsymbol{\theta}_j^*$
    **end**
    Find $\boldsymbol{\theta}^*$ such that

$$\log p(\widetilde{Y}_{(\cdot,s)}|\widetilde{U}, \boldsymbol{\theta}^*) \geq \log p(\widetilde{Y}_{(\cdot,s)}|\widetilde{U}, \boldsymbol{\theta}_j^*), \quad j = 1, \dots, n_{\mathrm{start}}$$

    /* Predicting */
    Compute $\mu_{\mathcal{D}_{i-1,k}}^{(s)}(\boldsymbol{U}_{i-1}^k)$ with (11) using $\boldsymbol{\theta}^*$
**end**
Set $\widehat{f}_{\mathrm{nn}}(\boldsymbol{U}_{i-1}^k) \leftarrow (\mu_{\mathcal{D}_{i-1,k}}^{(1)}(\boldsymbol{U}_{i-1}^k), \dots, \mu_{\mathcal{D}_{i-1,k}}^{(d)}(\boldsymbol{U}_{i-1}^k))^\top$

---

---

**Algorithm 3:** RandNets training procedure within RandNet-Parareal

---

**Input:** Input $\boldsymbol{U}_{i-1}^k$, dataset $\mathcal{D}_k$, number of neurons $M$, number of nearest neighbors $m_{\mathrm{RandNet}}$
**Output:** Prediction $\widehat{f}_{\mathrm{RandNet}}(\boldsymbol{U}_{i-1}^k)$ of $(\mathscr{F} - \mathscr{G})(\boldsymbol{U}_{i-1}^k)$

**Initialization**
Ensure each ODE/PDE coordinate takes values in $[-1, 1]$
/* Find the $m_{\mathrm{RandNet}}$ nns to $\boldsymbol{U}_{i-1}^k$, and compute the reduced dataset (10) */
$\mathcal{D}_{i-1,k} \leftarrow \{(\boldsymbol{U}_{\boldsymbol{U}_{i-1}^k}^{(l\text{-nn})}, \mathbf{Y}_{\boldsymbol{U}_{i-1}^k}^{(l\text{-nn})}),\ l = 1, \dots, m_{\mathrm{RandNet}}\} \subset \mathcal{D}_k$
Sample $A_{w,j} \sim \mathrm{Uniform}(-1, 1)$, $w = 1, \dots, M$, $j = 1, \dots, d$
Sample $\boldsymbol{\zeta}_w \sim \mathrm{Uniform}(-1, 1)$, $w = 1, \dots, M$
Let $\widetilde{X} \in \mathbb{R}^{m \times M}$

**Training**
$\widetilde{X}^\top \leftarrow \boldsymbol{\sigma}(A\widetilde{U}^\top + \boldsymbol{\zeta})$                              /* Using broadcasting on $\boldsymbol{\zeta}$ */
**if** $\mathrm{rank}(\widetilde{X}^\top \widetilde{X}) == M \leq m$ **then**
    $\widehat{W}^{\mathcal{D}_{i-1,k}} \leftarrow (\widetilde{X}^\top \widetilde{X})^{-1} \widetilde{X}^\top \widetilde{Y}$              /* Least-squares estimator */
**else**
    $\widehat{W}^{\mathcal{D}_{i-1,k}} \leftarrow (\widetilde{X}^\top \widetilde{X})^\dagger \widetilde{X}^\top \widetilde{Y}$

                                          /* Ridgeless interpolator */
**end**

**Predicting**
$\widehat{f}_{\mathrm{RandNet}}(\boldsymbol{U}_{i-1}^k) \leftarrow (\widehat{W}^{\mathcal{D}_{i-1,k}})^\top \boldsymbol{\sigma}(A\boldsymbol{U}_{i-1}^k + \boldsymbol{\zeta})$

---

# B  Additional details on the nnGParareal correction function

In this section, we provide more details on the nearest neighbors (nns) Gaussian process modeling, the mathematical expressions of the nnGParareal correction function $\widehat{f}_{\mathrm{nnGPara}}$, and the reduced dataset $\mathcal{D}_{i-1,k}$. These are not explicitly presented in the main text as they require additional notation, which we believe does not enrich the explanation. While the description of GPs presented here is for nnGParareal (and the corresponding nnGPs), it immediately generalizes to GParareal by replacing the reduced dataset $\mathcal{D}_{i-1,k}$ with the full dataset $\mathcal{D}_k$. The interested reader can find more details in the original papers, [21] and [57].

Let the set of inputs $\boldsymbol{U}_{i-1}^j \in \mathbb{R}^d$ and outputs $(\mathscr{F} - \mathscr{G})(\boldsymbol{U}_{i-1}^j) \in \mathbb{R}^d$, $i = 1, \ldots, N$, $j = 0, \ldots, k-1$, collected by iteration $k$, be denoted by $\mathcal{U}_k$ and $\mathcal{Y}_k$, respectively. Now, define $\mathcal{D}_{i-1,k}$ as the restriction of $\mathcal{D}_k$ to the $m$ nns of $\boldsymbol{U}_{i-1}^k$ in $\mathcal{U}_k$, namely

$$\mathcal{D}_{i-1,k} := \{(\boldsymbol{U}_{\boldsymbol{U}_{i-1}^k}^{(l\text{-nn})}, \mathbf{Y}_{\boldsymbol{U}_{i-1}^k}^{(l\text{-nn})}), \;\; l = 1, \ldots, m\} \subset \mathcal{D}_k,$$

where $\mathbf{Y}_{\boldsymbol{U}_{i-1}^k}^{(l\text{-nn})} = (\mathscr{F} - \mathscr{G})(\boldsymbol{U}_{\boldsymbol{U}_{i-1}^k}^{(l\text{-nn})}) \in \mathcal{Y}_k$, and $\boldsymbol{U}_{\boldsymbol{U}_{i-1}^k}^{(l\text{-nn})}$ is the $l$th nn of $\boldsymbol{U}_{i-1}^k$ in $\mathcal{D}_k$, i.e. the $l$th ordered statistics of the set formed out of Euclidean distances $\|\boldsymbol{U}_{i-1}^j - \boldsymbol{U}'\|$ between $\boldsymbol{U}_{i-1}^j$ and any $\boldsymbol{U}' \in \mathcal{U}_k$. That is, there exists $\boldsymbol{U}_1, \ldots, \boldsymbol{U}_l = \boldsymbol{U}_{\boldsymbol{U}_{i-1}^k}^{(l\text{-nn})} \in \mathcal{U}_k$ such that, for any $\boldsymbol{U}' \in \mathcal{U}_k, \boldsymbol{U}' \neq \boldsymbol{U}_r, r = 1, \ldots, l$, we have

$$\|\boldsymbol{U}_{i-1}^j - \boldsymbol{U}_1\| \leq \ldots \leq \|\boldsymbol{U}_{i-1}^j - \boldsymbol{U}_{l-1}\| \leq \|\boldsymbol{U}_{i-1}^j - \boldsymbol{U}_l\| \leq \|\boldsymbol{U}_{i-1}^j - \boldsymbol{U}'\|. \tag{10}$$

Finally, let $\widetilde{U}, \widetilde{Y} \in \mathbb{R}^{m \times d}$ be the matrices of input nns and outputs collected in $\mathcal{D}_{i-1,k}$, respectively.

In nnGParareal, following the Bayesian framework, a GP prior is placed over the correction function $\mathscr{F} - \mathscr{G}$ for each of the $d$ coordinates as

$$(\mathscr{F} - \mathscr{G})_s \sim GP(\mu_{\mathrm{GP}}^{(s)}, \mathcal{K}_{\mathrm{GP}}), \;\; s = 1, \ldots, d,$$

where $\mu_{\mathrm{GP}}^{(s)} : \mathbb{R}^d \to \mathbb{R}$ is the prior mean function, taken to be zero for all $s = 1, \ldots, d$, and $\mathcal{K}_{\mathrm{GP}} : \mathbb{R}^d \times \mathbb{R}^d \to \mathbb{R}$ is the exponential prior variance kernel function

$$\mathcal{K}_{\mathrm{GP}}(\boldsymbol{U}, \boldsymbol{U}') = \sigma_{\mathrm{o}}^2 \exp(-\|\boldsymbol{U} - \boldsymbol{U}'\|^2 / \sigma_{\mathrm{in}}^2),$$

with $\sigma_{\mathrm{in}}^2$ and $\sigma_{\mathrm{o}}^2$ denoting the input and output length scales, respectively. Differently from the prior mean, the prior variance is the same across the $d$ components. Then, each nnGParareal prediction $\widehat{f}_{\mathrm{nnGPara}}^{(s)}(\boldsymbol{U}_{i-1}^k) \in \mathbb{R}$, $s = 1, \ldots, d$, is obtained from the GP posterior mean $\mu_{\mathcal{D}_{i-1,k}}^{(s)}(\boldsymbol{U}_{i-1}^k) \in \mathbb{R}$, computed on the reduced dataset $\mathcal{D}_{i-1,k}$, given by

$$\widehat{f}_{\mathrm{nnGPara}}^{(s)}(\boldsymbol{U}_{i-1}^k) = \mu_{\mathcal{D}_{i-1,k}}^{(s)}(\boldsymbol{U}_{i-1}^k) := \mathcal{K}(\widetilde{U}, \boldsymbol{U}_{i-1}^k)^\top (\mathcal{K}(\widetilde{U}, \widetilde{U}) + \sigma_{\mathrm{reg}}^2 \mathbb{I}_m)^{-1} \widetilde{Y}_{(\cdot, s)}, \tag{11}$$

where $\mathcal{K}(\widetilde{U}, \boldsymbol{U}_{i-1}^k) \in \mathbb{R}^m$ is a vector of covariances between every input collected in $\widetilde{U}$ and $\boldsymbol{U}_{i-1}^k$ defined as $(\mathcal{K}(\widetilde{U}, \boldsymbol{U}_{i-1}^k))_r = \mathcal{K}_{\mathrm{GP}}((\widetilde{U}_{(r, \cdot)})^\top, \boldsymbol{U}_{i-1}^k)$, $r = 1, \ldots, m$, and $\mathcal{K}(\widetilde{U}, \widetilde{U}) \in \mathbb{R}^{m \times m}$ is the covariance matrix, with $(\mathcal{K}(\widetilde{U}, \widetilde{U}))_{q,r} = \mathcal{K}_{\mathrm{GP}}((\widetilde{U}_{(q, \cdot)})^\top, (\widetilde{U}_{(r, \cdot)})^\top)$, $r, q = 1, \ldots, m$. Here, $\sigma_{\mathrm{reg}}^2$ denotes a regularization term, also known as nugget, jitter, or regularization strength, which is added to improve the numerical stability when computing the inverse matrix, see [21] for further details. The hyperparameters $\boldsymbol{\theta} := (\sigma_{\mathrm{in}}^2, \sigma_{\mathrm{o}}^2, \sigma_{\mathrm{reg}}^2)$ entering into the posterior mean and prediction (11) control the performance of the GP, and are optimized by numerically maximizing the marginal log-likelihood:

$$\log p(\widetilde{Y}_{(\cdot, s)} | \widetilde{U}, \boldsymbol{\theta}) \propto -\widetilde{Y}_{(\cdot, s)}^\top (\mathcal{K}(\widetilde{U}, \widetilde{U}) + \sigma_{\mathrm{reg}}^2 \mathbb{I}_m)^{-1} \widetilde{Y}_{(\cdot, s)} - \log \det(\mathcal{K}(\widetilde{U}, \widetilde{U})), \tag{12}$$

where $\mathcal{K}(\cdot, \cdot)$ depends on $\boldsymbol{\theta}$ through the kernel $\mathcal{K}_{\mathrm{GP}}$, and $\det(A)$ denotes the determinant of a square matrix $A$. For a thorough treatment of Gaussian processes, including derivation of the likelihood and of the posterior distribution (which is Gaussian with mean as in (11), see [80].

# C  Computational complexity analysis

Consider the $d$-dimensional initial value problem (1) for some (O/P)DE. Let $N$ be the number of subintervals (data points) at each $k$th iteration of the PinT algorithm. For any $k$th iteration of

the scheme, a total of $Nk$ data points, each $d$-dimensional, are available. Here, we provide the computational cost of RandNet-Parareal, and compare it to that of nnGParareal, the state-of-the-art Parareal algorithm proposed in [21]. Both RandNet-Parareal and nnGParareal use only the reduced data set of $m$ nns to a given point to construct its image-prediction via (2). Note that the $m$ nns (in Euclidean distance) to some point $\boldsymbol{U} \in \mathbb{R}^d$ among $Nk$ available points are found at a cost which is at most linear in the sample size, that is $O(mNk)$ (for moderate dimensions $d$, one can get an improved cost $O(m\log(Nk))$, logarithmic in the sample size) [21]. Since our goal is to compare the computational complexities of nnGParareal and RandNet-Parareal as a function of $d$, we consider the worst-case complexity of the nns search.

Given an input $\boldsymbol{U}_{i-1}^k \in \mathbb{R}^d$, $i = 1, \ldots, N$ at iteration $k$, the computational model cost of a prediction $\boldsymbol{U}_{i-1}^k$ produced by all $d$ models of $m_{\mathrm{nnGP}}$-nnGPs at iteration $k$ via the predictor-corrector rule (2) with nnGParareal correction (11) and $m_{\mathrm{nnGP}}$ nns is given in [21] as

$$T_{\mathrm{nnGP}}(k) \leq C_{\mathrm{nnGP}} Nk (n_{\mathrm{start}} n_{\mathrm{reg}} \frac{d}{N} \vee 1) \times$$

$$(\underbrace{m_{\mathrm{nnGP}} d}_{B:=\mathcal{K}(U,\boldsymbol{U}_{i-1}^{k-1})^\top} + \underbrace{m_{\mathrm{nnGP}}^2 d}_{C:=\mathcal{K}(U,U)} + \underbrace{m_{\mathrm{nnGP}}^3}_{D:=(B+\sigma_{\mathrm{reg}}^2 \mathbb{I}_{m_{\mathrm{nnGP}}})^{-1}} + \underbrace{m_{\mathrm{nnGP}}^2}_{B \cdot D} + \underbrace{m_{\mathrm{nnGP}} d}_{BD \cdot Y} + \underbrace{m_{\mathrm{nnGP}} Nk}_{\text{nearest neighbors}})$$

$$= C_{\mathrm{nnGP}} Nk (n_{\mathrm{reg}} n_{\mathrm{start}} \frac{d}{N} \vee 1)(m_{\mathrm{nnGP}}^3 + m_{\mathrm{nnGP}}^2 + d(m_{\mathrm{nnGP}}^2 + 2m_{\mathrm{nnGP}}) + m_{\mathrm{nnGP}} Nk),$$

with $C_{\mathrm{nnGP}}$ being some constant that in general *does depend* on $k$, $m_{\mathrm{nnGP}}$, and $d$. Also, $n_{\mathrm{reg}}$ and $n_{\mathrm{start}}$ correspond to the number of random restarts and the number of explored values of the regularization penalty in the kernel regression (associated to the hyperparameter optimization (see [21, Section 4.5]), respectively. Furthermore, $\vee$ is the maximum operator, and the factor $(n_{\mathrm{start}} n_{\mathrm{reg}} d/N \vee 1) \geq 1$ follows from the fact that $d$ independent nnGPs and hyperparameter optimization are parallelized over the $N$ cores.

In RandNet-Parareal, the correction term $\widehat{f}_{\mathrm{RandNet\text{-}Para}}$ is modeled by the random weights neural network and evaluated as (8). Again, only $m_{\mathrm{RandNet}}$ nns (in Euclidean distance) to $\boldsymbol{U}_{i-1}^k$ are used to construct the prediction, leading to the following computational model cost at iteration $k$:

$$T_{\mathrm{RandNet}}(k) \leq C_{\mathrm{RandNet}} Nk \frac{1}{N} (\underbrace{Mdm_{\mathrm{RandNet}}}_{X:=\boldsymbol{\sigma}(A \cdot U + \boldsymbol{\zeta})} + \underbrace{M^2 m_{\mathrm{RandNet}}}_{\Sigma:=X \cdot X^\top} + \underbrace{Mr^2}_{\Sigma^\dagger}$$

$$+ \underbrace{Mm_{\mathrm{RandNet}} d}_{\Sigma^\dagger \cdot X} + \underbrace{M^2 d}_{W:=\Sigma^\dagger X \cdot Y} + \underbrace{Mdm_{\mathrm{RandNet}}}_{W^\top \cdot X} + \underbrace{m_{\mathrm{RandNet}} Nk}_{\text{nearest neighbors}})$$

$$= C_{\mathrm{RandNet}} k (Mr^2 + M^2 m_{\mathrm{RandNet}} + d(M^2 + 3Mm_{\mathrm{RandNet}}) + m_{\mathrm{RandNet}} Nk),$$

where $M$ is the number of hidden neurons, $r$ is the rank of the covariance of activated neurons $\Sigma$ (mind that the pseudoinverse of $\Sigma$ would contribute cubically in $m$ only if $\Sigma$ is of full rank numerically, which is not observed empirically) and $C_{\mathrm{RandNet}}$ is a constant *independent* on $N, k, M, d, m_{\mathrm{RandNet}}$. The factor $1/N$ in the first inequality corresponds to parallelization over $N$ processors.

We note the following differences in costs between these two algorithms according to realistic situations:

- $d \gg N$ in most relevant applications, especially for PDEs. Hence, $(n_{\mathrm{start}} n_{\mathrm{reg}} d/N \vee 1) \gg 1$, limiting the benefits from parallelization for nnGParareal. In the considered experiments, we had access to a maximum of approximately $N = 500$ processors, while we considered up to $d \approx 10^5$. It is easy to see that $T_{\mathrm{nnGP}}$ is quadratic in dimension $d$, while $T_{\mathrm{RandNet}}$ is only linear. This difference is mainly due to the factor $(n_{\mathrm{start}} n_{\mathrm{reg}} d/N \vee 1) \geq 1$ in $T_{\mathrm{nnGP}}$ as opposed to $1/N$ in $T_{\mathrm{RandNet}}$.

- Although $M > m_{\mathrm{nnGP}}$, $M = 100$ is sufficient for consistent performance across a range of systems, as shown in our numerical experiments.

- nnGParareal incurs additional cost due to hyperparameter optimization [21], necessary for tuning the kernel input and output scales and the regularization strength for each of the $d$ dimensions, which is performed by maximizing the loglikelihood. First, to explore the parameter space and allow for multiple starting points given the nonconvex optimization

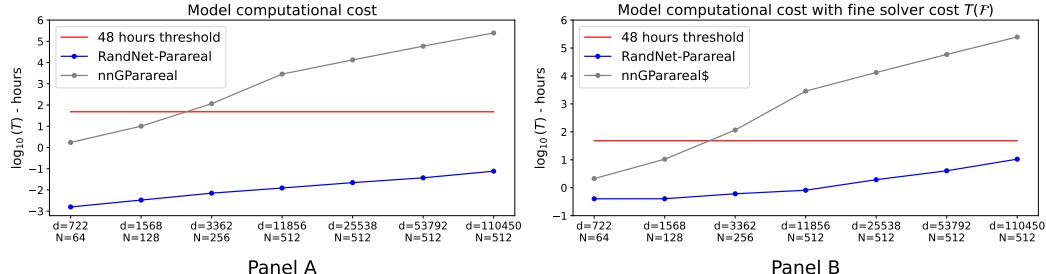

Figure 3: Theoretical *model* cost (panel A) and theoretical *total* cost (panel B), as functions of the dimension $d$ (and the corresponding $N$). The results are reported in terms of $\log_{10}(\text{hours})$.

problems, both $n_{\text{reg}}$ and $n_{\text{start}}$ should be set large. Second, each loglikelihood maximization conducted per dimension of the system requires a large number of iterations performed sequentially. Hence, $C_{\text{nnGP}} \gg C_{\text{RandNet}}$, with $C_{\text{nnGP}}$ depending, in general, on $k$, $m_{\text{nnGP}}$, $d$, as opposed to $C_{\text{RandNet}}$. Indeed, RandNet requires no tuning (a significant advantage with respect to GPs and nnGPs, making it more user-friendly), neither for the distribution of the random weights nor for the regularization parameter $\lambda$, due to the use of the ridgeless estimator. Empirically, we observed $C_{\text{nnGP}}/C_{\text{RandNet}}$ to be up to 1000 (this can be seen in Figure 1).

- We emphasize that since the ReLu function is chosen as activation in RandNet, the matrix $X$ of activated neurons is sparse with sparsity degree $\gamma$. Hence, the computational complexity $T_{\text{RandNet}}$ could be further improved, as the computational complexity of sparse operations is proportional to the number of nonzero elements in the matrix. We intentionally left these arguments out of the complexity analysis, since we do not use sparse operations in our code implementation.

- The upper bound of $T_{\text{RandNet}}$ could potentially be improved further, as additional parallelization may occur during standard matrix operations, depending on the specific computing environment.

Figure 3 illustrates the theoretical *model* costs $T_{\text{nnGP}}$ and $T_{\text{RandNet}}$ (Panel A) and theoretical *total* costs obtained by adding the coarse and fine solver costs (Panel B), as functions of the dimension $d$ (and the corresponding $N$). The results are reported in terms of $\log_{10}(\text{hours})$. To calibrate the constants in both complexity bounds, we used the total empirical computational cost in Figure 1, together with its breakdown described in Table 8. Panel A shows that RandNet-Parareal displays significant improvement in scalability with respect to the state-of-the-art Parareal algorithm nnG-Parareal, while Panel B demonstrates that whenever the cost of the fine solver is added, our results are in full coherence with the empirical results.

## D Robustness study

In this section, we study the robustness of RandNet-Parareal to changes in the number of nns $m_{\text{RandNet}}$, the number of neurons $M$, and the randomly sampled values of neural network weights $A, \zeta$. Our empirical findings (for two of the three considered PDEs) demonstrate that the iterations $K_{\text{RandNet-Para}}$ to convergence for RandNet-Parareal remain largely consistent despite variations in these factors. This ensures robust performance across a broad spectrum of parameter values, reducing users' need for extensive tuning. For computational tractability, we limit the robustness analysis to relatively small systems, such as Burgers' equation with $d = 128$, and the Diffusion-Reaction equation with $d = 722$, conducting 100 weight samplings for each system. For every set of weights, we iterate RandNet-Parareal across $m_{\text{RandNet}}$ values ranging from 2 to 20, and $M$ values ranging from 20 to 500 in increments of 10. The proportions of iterations needed to converge across 100 runs for different values of $m_{\text{RandNet}}$ and $M$ for the Burger's and diffusion-Reaction equations are reported in Figures 4 and 5, respectively. Although we observe some minor differences between the two systems, the main trend is clear: as long as reasonable values of $m_{\text{RandNet}}$ and $M$ are chosen, the iterations to convergence for RandNet-Parareal vary at most by a few units when changing the values of $m_{\text{RandNet}}$, $M$ or a particular sampling seed of weights. Nevertheless, larger $M$ might improve

the performance, since, in this case, RandNets operate in the interpolation regime, as discussed in Section 4.

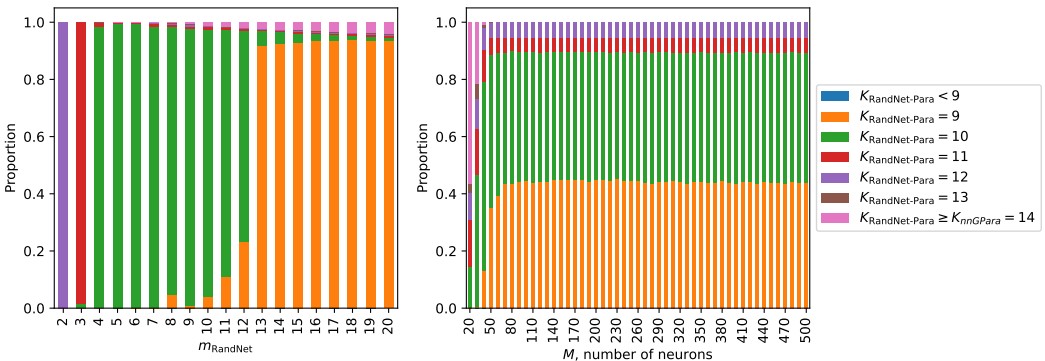

Figure 4: Histogram of the iterations to convergence $K_{\text{RandNet-Para}}$ of RandNet-Parareal for $d = 128$ for Burgers' equation. We sample the network weights $A$, $\boldsymbol{\zeta}$ 100 times. For each set of weights, we run RandNet-Parareal for $m_{\text{RandNet}} \in \{2, 3, \ldots, 20\}$ and $M \in \{20, 30, 40, \ldots, 500\}$. The left and right panels show the aggregated histograms of $K_{\text{RandNet-Para}}$ versus $m_{\text{RandNet}}$ and $M$, respectively.

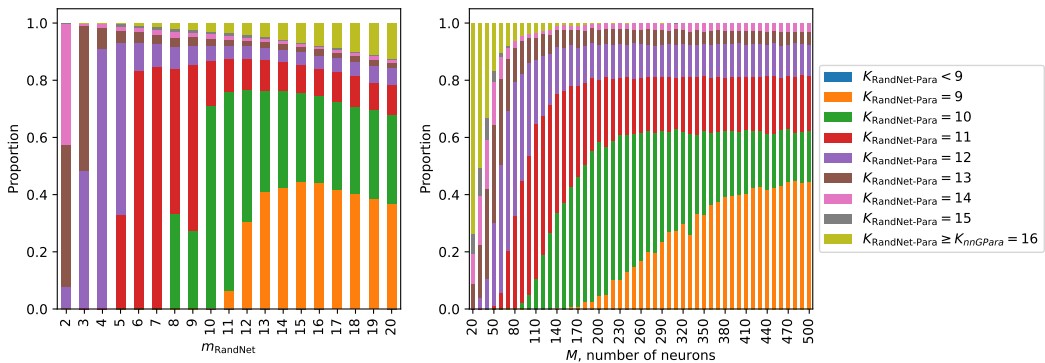

Figure 5: Histogram of the iterations to convergence $K_{\text{RandNet-Para}}$ of RandNet-Parareal for $d = 722$ for Diffusion-Reaction equation. We sample the network weights $A$, $\boldsymbol{\zeta}$ 100 times. For each set of weights, we run RandNet-Parareal for $m_{\text{RandNet}} \in \{2, 3, \ldots, 20\}$ and $M \in \{20, 30, 40, \ldots, 500\}$. The left and right panels show the aggregated histograms of $K_{\text{RandNet-Para}}$ versus $m_{\text{RandNet}}$ and $M$, respectively.

## E    Additional numerical experiments: 2D and 3D Brusselator PDE

Here, we carry out an additional scalability study for the 2 and 3 spatial dimensional Brusselator PDE. This model is a two-component reaction system that exhibits complex behavior, including oscillations, spatial patterns, and chaos. It is described by

$$\partial_t u = D_0 \nabla^2 u + a - (1 + b)u + vu^2,$$

and

$$\partial_t v = D_1 \nabla^2 v + bu - vu^2.$$

In chemistry, the components $u, v$ refer to the concentration of two substances, whereas the constants $D_0, D_1$ are the respective diffusivity of each component, indicating the rate at which the substances spread out in space. Moreover, the parameters $a$ and $b$ are related to reaction rates. In our experiments, we used $D_0 = 0.1$, $D_1 = 0.1D_0$, $a = 1$, and $b = 3$. We take $t \in [0, 35]$, $(u, v) \in (-1, 1)^2 \times (-1, 1)^2$ for the 2D Brusselator, and $(u, v) \in (-1, 1)^3 \times (-1, 1)^3$ for the three spatial dimension case. We initialize the $u$ values at time $t = 0$ by setting them equal to $a$, and the $v$ values by taking them normally distributed over the spatial grid. Further details regarding the number of spatial

Table 3: Simulation setup for the 2D and 3D Brusselator

| Domain | $N_u = N_v$ | $d$ | $\mathscr{G}$ | $\mathscr{G}_{\Delta t}$ | $\mathscr{F}$ | $\mathscr{F}_{\Delta t}$ | $N$ |
|---|---|---|---|---|---|---|---|
| $(u,v,t) \in (-1,1)^2 \times (-1,1)^2 \times [0,35]$ | 32 | 2048 | RK1 | 0.034 | RK4 | $1e^{-7}$ | 512 |
| $(u,v,t) \in (-1,1)^2 \times (-1,1)^2 \times [0,35]$ | 64 | 8192 | RK1 | 0.033 | RK4 | $1e^{-7}$ | 512 |
| $(u,v,t) \in (-1,1)^3 \times (-1,1)^3 \times [0,35]$ | 20 | 16000 | RK1 | 0.052 | RK4 | $1e^{-7}$ | 512 |
| $(u,v,t) \in (-1,1)^3 \times (-1,1)^3 \times [0,35]$ | 25 | 31250 | RK1 | 0.057 | RK4 | $1e^{-7}$ | 512 |

$N_u$ and $N_v$ are the number of spatial discretization points for $u$ and $v$ along each spatial dimension, yielding a $d = 2N_x^2$- or $d = 3N_x^3$-dimensional ODE, depending on the considered system. $\mathscr{G}$ and $\mathscr{F}$ denote the coarse and fine solvers, respectively, while the $_{\Delta t}$ subscript refers to the timestep. The number of nns used for $\mathcal{D}_{i-1,k}$ in nnGParareal and RandNet-Parareal are $m_{\mathrm{nnGP}} = 20$ and $m_{\mathrm{RandNet}} = 4$, respectively. $N$ is the total number of intervals.

discretizations, the number of intervals $N$ and the order of the solvers $\mathscr{F}$ and $\mathscr{G}$ is given in Table 3. Figure 6 highlights the strong scaling advantages of RandNet-Parareal compared to nnGParareal, setting $N = 512$ and restricting the runtime budget to a maximum of 48 hours, as done in the other test cases.

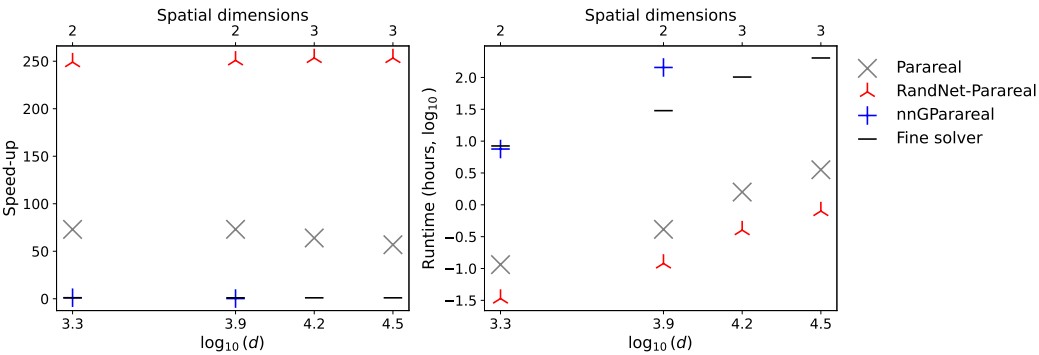

Figure 6: Scalability study for the 2 and 3 spatial dimensional Brusselator PDE. We used $N = 512$ and 48 hours runtime budget. nnGParareal for $\log_{10}(d) = 3.9$ is estimated, as the algorithm does not converge within the 48 hours runtime budget.

# F   Accuracy and runtimes across models and algorithms

In Table 4 below, we report the accuracies and runtimes (shown in parentheses) for RandNet-Parareal, Parareal, and nnGParareal. The accuracy is measured with maximum absolute error (mean across intervals) with respect to the true solution obtained by running $\mathscr{F}$ sequentially. Interestingly, all accuracies are far below the pre-defined accuracy level $\epsilon$, with RandNet-Parareal achieving the lowest one in all but one experiment, with much smaller runtimes across all case studies.

# G   Simulation setups

This section summarizes the simulation setups used for producing the results discussed in Section 5 in the main text. The tables below report the space and time domain of the considered PDEs, the number of spatial discretization points $N_x$ (and $N_y$, in case of two-dimensional spatial systems), the numerical solvers used for $\mathscr{G}$ and $\mathscr{F}$, their corresponding numbers of time steps per interval, the number of intervals $N$, and the number of nns used for nnGParareal ($m_{\mathrm{nnGP}}$) and RandNet-Parareal ($m_{\mathrm{RandNet}}$). In particular, Table 5 refers to the viscous Burgers' equation, Table 6 to the Diffusion-Reaction equation, and Table 7 to the shallow water equations (SWEs).

Table 4: Accuracy and computational cost of the three considered algorithms

| PDE | RandNet-Parareal | Parareal | nnGParareal |
|---|---|---|---|
| Burgers' $d = 128$ | $1.06e^{-8}$ (1h 2m) | $1.85e^{-8}$ (8h 54m) | $1.32e^{-7}$ (1h 39m) |
| Diffusion-Reaction $d = 7.2e^2$ | $3.56e^{-8}$ (23m) | $1.83e^{-8}$ (1h 40m) | $5.71e^{-7}$ (1h 11m) |
| Diffusion-Reaction $d = 3.3e^3$ | $8.56e^{-10}$ (33m) | $2.45e^{-8}$ (7h 52m) | not converged |
| Diffusion-Reaction $d = 2.5e^4$ | $8.09e^{-11}$ (1h 57m) | $7.43e^{-9}$ (9h 50m) | not converged |
| SWE $d = 3.1e^4$ | $6.75e^{-8}$ (4h 9m) | $5.15e^{-8}$ (15h 43m) | not converged |
| SWE $d = 6.1e^4$ | $8.54e^{-9}$ (12h 34m) | $2.84e^{-8}$ (19h 30m) | not converged |
| Brusselator 2D $d = 2e^3$ | $2.09e^{-8}$ (2m) | $3.16e^{-8}$ (7m) | $3.38e^{-7}$ (7h 31m) |

Accuracy and computational cost comparison of RandNet-Parareal, Parareal, and nnGParareal for different PDEs, with runtimes reported in parentheses. The accuracy is measured as maximum absolute error (mean across intervals) with respect to $\mathscr{F}$ run sequentially.

Table 5: Simulation setup for the viscous Burgers' equation

| Domain | $N_x$ | $d$ | $\mathscr{G}$ | $N_{\mathscr{G}}$ | $\mathscr{F}$ | $N_{\mathscr{F}}$ | $N$ | $m_{\mathrm{nnGP}}$ | $m_{\mathrm{RandNet}}$ |
|---|---|---|---|---|---|---|---|---|---|
| $(x,t) \in [-1,1] \times [0,5.9]$ | 128 | 128 | RK1 | 4 | RK8 | $4e^4$ | 128 | 18 | 3 |
| $(x,t) \in [-1,1] \times [0,5.9]$ | 1128 | 1128 | RK1 | 293 | RK8 | $6e^5$ | 128 | 18 | 3 |

$N_x$ is the number of space discretizations, the same as $d$ here. $\mathscr{G}$ and $\mathscr{F}$ denote the chosen coarse and fine solvers, with corresponding time discretization steps per interval $N_{\mathscr{G}}$ and $N_{\mathscr{F}}$, respectively. Here $N$ is the number of intervals, while $m_{\mathrm{nnGP}}$ and $m_{\mathrm{RandNet}}$ are the numbers of nns used to create $\mathcal{D}_{i-1,k}$ for nnGParareal and RandNet-Parareal, respectively.

### G.1 Simulation setup for the SWEs

Here, we give more details on the radial dam break simulation of Section 5.3. Our domain consists of a rectangular box defined as $(x,y) \in [-5,5] \times [0,5]$, which we evolve temporally over $t \in [0,20]$. Following [75], as an initial condition, we place a Gaussian-shaped column of water centered at $(x,y) = (-2.5, 1.5)$, with covariance matrix $\Sigma = \begin{pmatrix} 0.25 & 0 \\ 0 & 0.25 \end{pmatrix}$. We use Neumann boundary conditions, and evolve the system using $N = 235$ intervals over four increasingly finer spatial meshes, as described in Table 7. We used the *ParareaML* [8] Python package to implement the SWEs and corresponding numerical solvers.

Table 6: Simulation setup for the Diffusion-Reaction equation

| Domain | $N_x$ | $N_y$ | $d$ | $\mathscr{G}$ | $N_{\mathscr{G}}$ | $\mathscr{F}$ | $N_{\mathscr{F}}$ | $N$ |
|---|---|---|---|---|---|---|---|---|
| $(x,y,t) \in [-1,1]^2 \times [0,20]$ | 19 | 19 | 722 | RK1 | 1 | RK4 | NA | 64 |
| $(x,y,t) \in [-1,1]^2 \times [0,20]$ | 28 | 28 | 1568 | RK1 | 1 | RK4 | NA | 128 |
| $(x,y,t) \in [-1,1]^2 \times [0,20]$ | 41 | 41 | 3362 | RK1 | 1 | RK4 | NA | 256 |
| $(x,y,t) \in [-1,1]^2 \times [0,20]$ | 77 | 77 | 11858 | RK4 | 1 | RK8 | NA | 512 |
| $(x,y,t) \in [-1,1]^2 \times [0,20]$ | 113 | 113 | 25538 | RK4 | 2 | RK8 | NA | 512 |
| $(x,y,t) \in [-1,1]^2 \times [0,20]$ | 164 | 164 | 53792 | RK4 | 4 | RK8 | NA | 512 |
| $(x,y,t) \in [-1,1]^2 \times [0,20]$ | 235 | 235 | 110450 | RK4 | 8 | RK8 | NA | 512 |

$N_x$ and $N_y$ are the number of spatial discretization points for $x$ and $y$, respectively, yielding a $d = 2N_xN_y$-dimensional ODE. $\mathscr{G}$ and $\mathscr{F}$ denote the coarse and fine solvers, respectively. The number of nns used for $\mathcal{D}_{i-1,k}$ in nnGParareal and RandNet-Parareal are $m_{\mathrm{nnGP}} = 20$ and $m_{\mathrm{RandNet}} = 3$, respectively. $N_{\mathscr{G}}$ is the time discretization steps of $\mathscr{G}$ per interval. $N_{\mathscr{F}} = $ NA since $\mathscr{F}$'s step size is chosen by *scipy* Runge-Kutta method [77].

Table 7: Simulation setup for the SWEs

| Domain | $N_x$ | $N_y$ | $d$ | $\mathscr{G}$ | $N_{\mathscr{G}}$ | $\mathscr{F}$ | $N_{\mathscr{F}}$ | $N$ |
|---|---|---|---|---|---|---|---|---|
| $(x,y,t) \in [-5,5] \times [0,5] \times [0,20]$ | 101 | 51 | 15453 | RK1 | 7 | RK4 | $1e^5$ | 235 |
| $(x,y,t) \in [-5,5] \times [0,5] \times [0,20]$ | 144 | 72 | 31104 | RK1 | 8 | RK4 | $2e^5$ | 235 |
| $(x,y,t) \in [-5,5] \times [0,5] \times [0,20]$ | 201 | 101 | 60903 | RK1 | 14 | RK4 | $4e^5$ | 235 |
| $(x,y,t) \in [-5,5] \times [0,5] \times [0,20]$ | 264 | 133 | 105336 | RK1 | 24 | RK4 | $5e^5$ | 235 |

$N_x$ and $N_y$ are the number of spatial discretization points for $x$ and $y$, respectively, leading to an ODE of dimension $d = 3N_xN_y$. $\mathscr{G}$ and $\mathscr{F}$ denote the chosen numerical coarse and fine solvers, respectively, with $N_{\mathscr{G}}$ and $N_{\mathscr{F}}$ being their corresponding time discretization steps per interval. In all cases, we set the number of nns used to create $\mathcal{D}_{i-1,k}$ to $m_{\mathrm{nnGP}} = 20$ for nnGParareal, and $m_{\mathrm{RandNet}} = 3$ for RandNet-Parareal.

## H Illustration of some PDE solutions

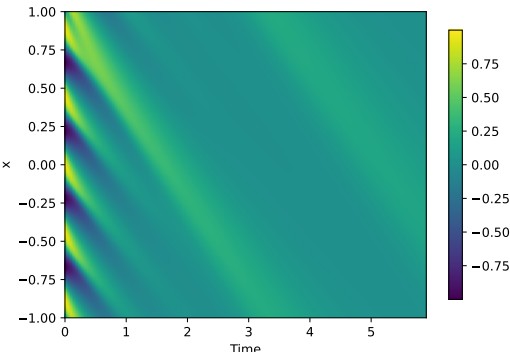

Figure 7: Numerical solution of viscous Burgers' equation over $(x, t) \in [-1, 1] \times [0, 5.9]$ with $d = 1128$ and initial conditions and additional settings as described in Section 5.1.

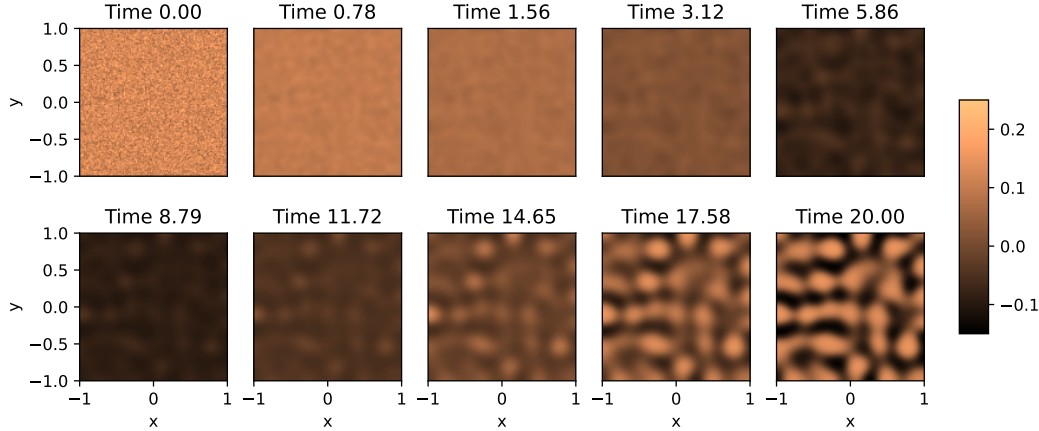

Figure 8: Numerical solution of the Diffusion-Reaction equation over $(x, y) \in [-1, 1]^2$ with $N_x = N_y = 235$ for a range of system times $t$. Only the activator $u(t, x, y)$ is plotted. The initial conditions and additional settings are as described in Section 5.2.

## I Additional simulation results for the Diffusion-Reaction equation

Here, we complement the results of the speed-ups and wallclock times reported in Figure 1 in the main text, with a detailed breakdown of the number of iterations to convergence, the runtimes of the coarse and fine solvers, the overall cost of training the model (up to convergence), and the total runtime, reported in Table 8.

Table 8: Speed-up analysis for the Diffusion-Reaction equation

| $d = 722, N = 64$ | | | | | | |
|---|---|---|---|---|---|---|
| Algorithm | $K$ | $NT_{\mathscr{G}}$ | $T_{\mathscr{F}}$ | $T_{\text{model}}$ | $T_{\text{alg}}$ | $S_{\text{alg}}$ |
| Fine | – | – | – | – | 2h 2m | 1 |
| Parareal | 53 | 0s | 2m | 0s | 1h 40m | 1.22 |
| nnGParareal | 16 | 0s | 2m | 40m | 1h 11m | 1.72 |
| RandNet-Parareal | 12 | 0s | 2m | 1s | 23m | **5.36** |

| $d = 1568, N = 128$ | | | | | | |
|---|---|---|---|---|---|---|
| Algorithm | $K$ | $NT_{\mathscr{G}}$ | $T_{\mathscr{F}}$ | $T_{\text{model}}$ | $T_{\text{alg}}$ | $S_{\text{alg}}$ |
| Fine | – | – | – | – | 4h 21m | 1 |
| Parareal | 93 | 0s | 2m | 0s | 3h 3m | 1.42 |
| nnGParareal | 20 | 0s | 2m | 10h 32m | 11h 12m | 0.39 |
| RandNet-Parareal | 12 | 0s | 2m | 4s | 25m | **10.26** |

| $d = 3362, N = 256$ | | | | | | |
|---|---|---|---|---|---|---|
| Algorithm | $K$ | $NT_{\mathscr{G}}$ | $T_{\mathscr{F}}$ | $T_{\text{model}}$ | $T_{\text{alg}}$ | $S_{\text{alg}}$ |
| Fine | – | – | – | – | 10h 58m | 1 |
| Parareal | 195 | 0s | 2m | 0s | 7h 52m | 1.40 |
| RandNet-Parareal | 12 | 0s | 3m | 20s | 33m | **19.87** |

| $d = 11858, N = 512$ | | | | | | |
|---|---|---|---|---|---|---|
| Algorithm | $K$ | $NT_{\mathscr{G}}$ | $T_{\mathscr{F}}$ | $T_{\text{model}}$ | $T_{\text{alg}}$ | $S_{\text{alg}}$ |
| Fine* | – | – | – | – | 2d 16h | 1 |
| Parareal | 58 | 1s | 7m | 0s | 6h 59m | 9.23 |
| RandNet-Parareal | 6 | 2s | 8m | 1m | 49m | **78.44** |

| $d = 25538, N = 512$ | | | | | | |
|---|---|---|---|---|---|---|
| Algorithm | $K$ | $NT_{\mathscr{G}}$ | $T_{\mathscr{F}}$ | $T_{\text{model}}$ | $T_{\text{alg}}$ | $S_{\text{alg}}$ |
| Fine* | – | – | – | – | 7d 16h | 1 |
| Parareal | 27 | 8s | 22m | 0s | 9h 50m | 19.25 |
| RandNet-Parareal | 5 | 9s | 23m | 2m | 1h 57m | **97.40** |

| $d = 53792, N = 512$ | | | | | | |
|---|---|---|---|---|---|---|
| Algorithm | $K$ | $NT_{\mathscr{G}}$ | $T_{\mathscr{F}}$ | $T_{\text{model}}$ | $T_{\text{alg}}$ | $S_{\text{alg}}$ |
| Fine* | – | – | – | – | 21d 7h | 1 |
| Parareal | 19 | 36s | 1h 0m | 0s | 19h 13m | 26.60 |
| RandNet-Parareal | 4 | 42s | 60m | 4m | 4h 6m | **124.87** |

| $d = 110450, N = 512$ | | | | | | |
|---|---|---|---|---|---|---|
| Algorithm | $K$ | $NT_{\mathscr{G}}$ | $T_{\mathscr{F}}$ | $T_{\text{model}}$ | $T_{\text{alg}}$ | $S_{\text{alg}}$ |
| Fine* | – | – | – | – | 56d 2h | 1 |
| Parareal | 14 | 3m | 2h 38m | 1s | 1d 14h | 35.84 |
| RandNet-Parareal | 4 | 3m | 2h 37m | 7m | 10h 48m | **124.52** |

Simulation study on the empirical scalability and speed-up of Parareal, nnGParareal (with $m_{\text{nnGP}} = 20$), and RandNet-Parareal (with $m_{\text{RandNet}} = 4$ and $M = 100$) for the Diffusion-Reaction equation. $T_{\mathscr{F}}$ and $T_{\mathscr{G}}$ refer to the runtimes per interval of the fine and coarse solvers, respectively, while $NT_{\mathscr{G}}$ is the runtime of the coarse solver over $N$ intervals. $T_{\text{model}}$ corresponds to the overall time to evaluate $\widehat{f}$, including training and predicting, until convergence at iteration $K$. $T_{\text{alg}}$ is the total algorithm runtime, while $S_{\text{alg}}$ is the parallel speed-up. "Fine*" indicates that the total runtime has been *estimated* extrapolating data from the other algorithms. Missing nnGParareal rows for $d \geq 3362$ are due to convergence failure within a 48-hour time budget.

