# OpenReview forum: "RandNet-Parareal: a time-parallel PDE solver using Random Neural Networks"
_NeurIPS.cc/2024/Conference — NeurIPS 2024 poster_

### Official Review · Reviewer_awZW · 2024-07-07

**Soundness:** 3
**Presentation:** 3
**Contribution:** 2
**Rating:** 6
**Confidence:** 2

**Summary:**

This paper proposes a new method for sequentially predicting and correcting numerical simulations by introducing random neural networks. The RandNets are single-layer feed-forward neural networks and only the output layer is for training. The numerical experiments have shown the improved training efficiency and scalability of the proposed method compared to the baseline models.

**Strengths:**

- This paper is well-written and well-organized. The theoretical guarantee of RandNets-Parareal is provided.

- Different types of PDEs have been used to evaluate the model performance.

**Weaknesses:**

- The studied problem is domain-specific, which might not be of general interest to the scientific machine-learning community.

- In Sec 5 Numerical Experiments, this paper provides the comparison of speed-ups/runtimes. It would be better to also show the varying solution accuracy with different setups.

- Sections 2 and 3 might be shortened and some of these parts can be moved to the Appendix. The Robustness study (Appendix C) can be moved to the main text since it highlights the benefits of the proposed method.

**Questions:**

In lines 7-8, is it a character "x" or a symbol "\times" for `x125`? Probably a symbol is preferred.

**Limitations:**

N/A.

---

> ### Author Rebuttal · Authors · 2024-08-07
>
> We would like to thank you for recognizing that our method shows improved training efficiency and scalability compared to the baseline methods. We also appreciate you acknowledge that our paper is well-written and well-organized, contains theoretical developments, and exemplifies RandNet-Parareal on different types of PDEs.
>
> In the paragraphs below, we replied in detail to all your commented weaknesses (W). We hope we can convince you that the paper merits a higher score, especially in view of our explanations of the importance of advancing the state-of-art on (parallel-in-time, PinT) numerical solvers and the implications of our work for the scientific computing community. We also advise you to look at the additional theoretical results (computational complexity, Tbl. A [pdf]) and empirical evidence (two more challenging examples, 2D and 3D Brusselator PDE, Fig. C [pdf]) we added in response to other referees' comments and questions during the time of the rebuttal, which we will incorporate in the camera-ready version of the paper.
>
> **W1: The studied problem is domain-specific, which might not be of general interest to the scientific machine-learning community.**
> **A**: We divide our answer into the following two parts:
> - _Importance of PinT schemes for scientific machine learning (ML)/computing and science in general_: Time and space efficient solving of (O/P)DEs remains one of the most important research directions of science and engineering. Broader availability of more affordable multicore high-performance computing resources makes parallelization more attractive. Much interest in the literature is concentrated on new numerical schemes, which allow for sub-linear in-time parallelization surpassing slow sequential solvers, especially crucial for long time horizons. Innovation in the area of PinT schemes is actively supported by research funding agencies (e.g. the ExCALIBUR cross-cutting research project "Exposing parallelism: Parallel in Time" involving the Met Office and UKAEA). Our proposed PinT algorithm leverages approximation and generalization properties of a flexible ML random NNs class, advancing the state of research on numerical Parareal solvers. Our work matches the NeurIPS topic of Primary Area on "Machine learning for physical sciences", opens the door to future research on randomized PinT numerical methods, and has implications for scientific computing, physics, and beyond [20,21,24]. Potential extensions of RandNet-Parareal include solvers for SDEs, coupled ODEs, and PDEs with noisy boundary conditions. We pursue some of these directions in our ongoing work.
>
> - _Scientific PinT computing for ML_: efficient ODE solvers started to gain much attention in the context of Neural ODEs, which are used as continuous-time models of certain NNs [22]. Other notable examples of ML models where PinT solvers could be game-changers are diffusion and generative models [23], for which the developed dedicated solvers could be further parallelized in time.
>
> **W2: In Sec. 5 Numerical Experiments, this paper provides the comparison of speed-ups/runtimes. It would be better to also show the varying solution accuracy with different setups.**
> **A**:  Thank you for the question, similar to Q3 of Ref FREq and W1 of Ref Eu5i. The fine solver $\mathcal{F}$, chosen by the user, defines the accuracy of the competing methods. All Parareal PinT schemes target the solution provided by $\mathcal{F}$. For any given $\epsilon>0$, all the solutions for all converged algorithms are $\epsilon$-close to the solution of $\mathcal{F}$ (eq. (4) [paper]). $\epsilon=5e^{-7}$ is set for all the examples considered.
>
> _Empirical evidence_: In the table below (as in our response to Q3 of Ref FREq), we report the accuracies and runtimes (shown in parentheses) for RandNet-Parareal, Parareal, and nnGParareal. The accuracy is measured as max. abs. error (mean across intervals) w.r.t. $\mathcal{F}$ (run sequentially). RandNet-Parareal has the best accuracy-cost trade-off, and one can easily see that under the same time budget for all approaches, RandNet-Parareal achieves higher accuracies than benchmarks.
>
> |PDE|RandNet-Parareal|Parareal|nnGParareal|
>  |- | -| -|- |
>  |Burgers' $d=128$|$1.06e^{-8}$ (1h 2m)|$1.85e^{-8}$ (8h 54m)|$1.32e^{-7}$ (1h 39m)|
>  |Diffusion-Reaction $d=7.2e^2$| $3.56e^{-8}$ (23m)| $1.83e^{-8}$ (1h 40m) |$5.71e^{-7}$ (1h 11m)|
>  |Diffusion-Reaction $d=3.3e^3$|$8.56e^{-10}$ (33m)| $2.45e^{-8}$ (7h 52m)|not converged|
>  |Diffusion-Reaction $d=2.5e^4$|$8.09e^{-11}$ (1h 57m)| $7.43e^{-9}$ (9h 50m)|not converged|
>  |SWE $d=3.1e^4$| $6.75e^{-8}$ (4h 9m)| $5.15e^{-8}$ (15h 43m)|not converged|
>  |SWE $d=6.1e^4$| $8.54e^{-9}$ (12h 34m)| $2.84e^{-8}$ (19h 30m)|not converged|
>
> To address comments from other Referees, we derived the theoretical complexity of RandNets and compared it to nnGParareal (see Table A [pdf]). Figs. A-B [pdf] plot the theoretical _model_ and _total_ costs (in $\log_{10}$(hours)), respectively, across dimension $d$ (and cores/subintervals $N$). To calibrate the constants in complexity bounds, we used the total empirical cost in Fig. 1 [paper] and its breakdown in Tbl. 6 [paper]. Fig. A shows significantly superior scalability of RandNet-Parareal w.r.t. nnGParareal. Fig. B shows that with the cost of $\mathcal{F}$ added, our results are fully consistent with those presented in the paper and in the table in our response to Q3 of Ref FREq.
>
> **W3: Sections 2 and 3 might be shortened and some of these parts can be moved to the Appendix. The Robustness study (Appendix C) can be moved to the main text since it highlights the benefits of the proposed method.**
> **A**: Thank you for this reasonable and nice suggestion. We hope to have a chance to introduce this change in the camera-ready version of the paper.
>
> **Minor Q. In lines 7-8, is it a character ``x'' or a symbol ``$\times$'' for x125? Probably a symbol is preferred.**
> **A**: Thank you for mentioning this. We will correct it accordingly.

---

> > ### Comment · Reviewer_awZW · 2024-08-10
> > **Response for rebuttal**
> >
> > Thanks for your rebuttal, especially the response to **W2**.  I still have some reservations about the significance of the work. I will be maintaining my score.

---

> ### Author Response · Authors · 2024-08-11
> **Follow-up to Reviewer awZW's comment**
>
> Thanks for your comment on our rebuttal and W2.
>
> We would appreciate if you could read what we wrote as a comment to Reviewer FREq about the impact of our paper on AI, which we report below for your convenience. We hope tis may reduce some of your reservations on the significance of our work. Properly communicating the relevance and impact of our method is very important for us.
>
> Our work exemplifies an instance when ML methods allow the advancement of various fields of science and engineering where solving (O/P)DEs is needed, see also our response to Ref FREq to his further question. We mentioned these applications in our initial rebuttal as our submission belongs to the Primary Area of *ML for Physical Sciences*.
>
> However, AI is also one of the fields that could greatly benefit from progresses regarding efficient ODE solvers, and thus our method. This is why we gladly provide examples of how our approach can also assist ML and AI, adding these points in the camera-ready version of the paper.
>
> ODEs are a crucial building block of some relevant techniques, such as *Diffusion models (DMs)*, *Neural ODEs*, *Optimal control and reinforcement learning*, and *Optimization for ML models*, to name a few. Below, we provide an example of how our work could lead to substantial advancements for DMs.
>
> - _Solvers for DMs_: In the context of DMs, the continuous time reversal process can be described by a probability flow ODE, defined by the score function, usually approximated with deep (convolutional) NNs (note that the ODE formulation offers significant advantages over SDEs in high dimensions). Faster ODE solvers can improve sampling speed, yielding faster image synthesis. The main existing directions in the literature involve using classical sequential solvers [30], developing faster and dedicated ones such as DDIM [24], DDPM [29], DPM-Solver [25], Heun [26], and parallelization of the autoregressive sampling process of DMs (by using Picard-Lindelöf iterations for ODE/SDE) [27,28]. The latter emerged mainly due to the need for other solutions to avoid the bottlenecks of sequential solvers. To the best of our knowledge, modulo this particular parallelized sampling, PinT schemes have not yet been used in this context. **This is the gap this paper could fill, as the existing successful sequential dedicated solvers could be embedded into our proposed RandNet-Parareal**. We expect this to be straightforward, since the goal is to collect samples via solving the corresponding (diffusion) ODEs at $[0,T]$, where RandNet-Parareal immediately finds its purpose.
>
> We refer to our latest comment to Reviewer FREq for an additional discussion on the GPU implementation which could further improve the efficiency of our method.
>
> We hope these clarifications provide you with evidence of the expected impact/implications of our method in the context of generative AI models and beyond.
>
> **References used above:**
> [24] J. Song, C. Meng, S. Ermon. Denoising diffusion implicit models. 2020.
> [25] C. Lu, Y. Zhou, F. Bao, J. Chen, C. Li, J. Zhu. DPM-solver: A fast ODE solver for diffusion probabilistic model sampling in around 10 steps. 2022.
> [26] T. Karras, M. Aittala, T. Aila, S. Laine. Elucidating the design space of diffusion-based generative models. 2022.
> [27] A. Shih, S. Belkhale, S. Ermon, D. Sadigh, N. Anari. Parallel sampling of diffusion models. 2023.
> [28] Z. Tang, J. Tang, H. Luo, F. Wang, T.-H. Chang. Accelerating parallel sampling of diffusion models. 2024.
> [29] J. Ho, A. Jain, P. Abbeel. Denoising diffusion probabilistic models. 2020.
> [30] Y. Song, J. Sohl-Dickstein, D.P. Kingma, A. Kumar, S. Ermon, B. Poole. Score-based generative modeling through stochastic differential equations. 2021.

---

### Official Review · Reviewer_nFhC · 2024-07-08

**Soundness:** 2
**Presentation:** 3
**Contribution:** 3
**Rating:** 6
**Confidence:** 3

**Summary:**

This paper proposes a method to accelerate the simulation of partial differential equations (PDEs) by converting them into systems of ordinary differential equations (ODEs). It then utilizes a framework that merges the random neural network and the parareal approach, termed RandNet-Parareal. For validation, three complex systems of PDEs are considered: the Burgers equation, the diffusion-reaction equation, and the shallow water equation. Results show gains achieved in terms of computational cost compared to other methods.

**Strengths:**

1. The proposed method accelerates neural PDE simulations by combining random neural networks with parareal concepts.

2. The considered PDEs are prototypical and complicated and exhibit the efficacy of the proposed approach.

**Weaknesses:**

1. The authors have not mentioned the trade-off between accuracy and computational cost, which is essential for high-fidelity simulations.

2. State-of-the-art comparisons are lacking, restricting the evaluation of the proposed method with random neural network-based PDE solvers.

3. The method is based on conventional numerical solvers, and hence, extending its applicability to complex geometries in higher dimensions may suffer from the curse of dimensionality.

4. It is often the case that random neural networks do not scale well to deep networks, and not much gain is achieved when using a deep network. The paper does not discuss extending the method to deep neural networks and what gains could be achieved.

**Questions:**

1. The proposed method's performance and compared methods are not quantified through a metric. It would be interesting to see the trade-off between accuracy and computational cost.

2. The method is not compared with other neural PDE solvers, specifically random neural network-based PDE solvers aiming to accelerate PDE simulations. Assessing the gain achieved compared to the methods proposed in the literature would be interesting.

3. As the method depends on traditional mesh-based numerical solvers, can the authors provide insights on how the method would perform in complicated geometries, particularly in higher dimensions?

4. Can the authors comment on the performance of their method when using a deep network instead of a shallow network?

**Limitations:**

The authors have mentioned the limitations of the work related to coarse numerical solvers and stiff systems. There are no social or ethical issues concerning the paper.

---

> ### Author Rebuttal · Authors · 2024-08-07
>
> We would like to thank you for acknowledging that our method succeeds in accelerating the solutions of complex PDEs. We appreciate you recognize the efficacy of our approach and see no limitations beyond those we mentioned in the paper.
>
> In the paragraphs below, we replied in detail to your questions (Q) and weaknesses (W) grouped by topic. We hope to convince you that our paper merits a higher score, as it pushes the boundaries of numerical solvers in the field of scientific computing by proposing a new direction of randomized numerical parallel-in-time (PinT) (O/P)DE solvers.
>
> **W1 & Q1: Discussion of the trade-off between accuracy (w.r.t. some metric) and computational cost, essential for high-fidelity simulations**
> **A**: Thank you for the question similar to Q3 of Ref FREq and W1 of Ref Eu5i. The fine solver _F_, chosen by the user, defines the accuracy of the competing methods. All Parareal PinT schemes target the solution provided by _F_. For any given $\epsilon>0$, all the solutions for all converged algorithms are $\epsilon$-close to the solution of _F_ (see eq.(4) in the paper, where another accuracy metric criterion could be used).
>
> _Empirical evidence_: In the table in our response to Q3 of Ref FREq (not copied here due to the space limit), we report the accuracies and runtimes for RandNet-Parareal, Parareal, and nnGParareal. Max. abs. error (mean across intervals) w.r.t. _F_ (run sequentially) is reported with the runtimes in parentheses. $\epsilon=5e^{-7}$ is chosen for the examples in the paper. One can clearly see that RandNet-Parareal has the best accuracy-cost trade-off.
>
> _Theoretical results_: To further address your comments and W1 of Ref Eu5i on the computational costs, we derived the theoretical complexity of RandNets and compared it to nnGParareal (see pdf for the details). Figs. A-B plot the theoretical _model_ cost and _total_ cost (in $\log_{10}$(hours)), respectively, across dimension $d$ (and cores/subintervals $N$). To calibrate the constants in both complexity bounds, we used the total empirical cost in Fig. 1 and its breakdown in Tbl. 6 [paper]. Fig. A shows significantly superior scalability of RandNet-Parareal w.r.t. nnGParareal, and Fig. B shows that once the cost of _F_ is added, our results are fully coherent with the empirical results in the paper and the table in our response to Q3 of Ref FREq.
>
> **W2 & Q2: Comparison with other state-of-the-art and neural PDE solvers, specifically RandNets-based ones, and assessment of gains achieved.**
> **A**: We kindly refer you to our extensive responses to similar questions Q1 of Ref FREq and W2 & Q2 of Ref Eu5i, which we cannot repeat here due to the space limits. We detailed why we focus exclusively on PinT (i.e., Parareal, nnGParareal) and not sequential (neural PDE) benchmarks. Importantly, RandNet-Parareal is the first PinT scheme in general, and Parareal algorithm in particular, that exploits random NNs, so no other NN-based PinT benchmark could be included in our comparative study. As the RandNet-Parareal construction is valid for any fine solver, any (neural) (O/P)DE solver can be readily incorporated as _F_ WLOG.
>
> **W3 & Q3: The method is based on mesh-based numerical solvers. Comment on applicability to complex geometries in higher dimensions (curse of dimensionality (CoD)).**
> **A**: We split the answer to this interesting question as follows:
>
> - _CoD in the learning_: Our goal is to approximate correction functions in high dimensions. NN functions are universal approximators (UA) (for example, they are dense in the set of continuous functions, defined on compacta, in the sup norm). Random NNs with random inner weights are proven to be UA of functions of certain regularity (related to Radon-wavelet integral representation in [4]; see more general setup in [12]). See also our reply to your W4 & Q4 below. Notably, under specific regularity assumptions on the correction function, it is possible to establish whether RandNets suffer or not from CoD (i.e., $M$ needs to grow exponentially in $d$ to maintain the same approximation accuracy) [4]. This opens avenues for provable (potentially not prone to CoD) speed and uncertainty quantification improvements with RandNet-Parareal, some of which we pursue in our ongoing work.
> - _CoD in the cost_: Albeit numerous attempts, the state-of-the-art Parareal algorithms still suffer from unfavourable on-dimension cost dependence. We argue in our answer to W1 of Ref Eu5i and our answer to your W1 & Q1 that our main competitor, nnGParareal, recently proposed in [1] to reduce the GP training cost, also has this issue (see pdf), while RandNet-Parareal needs no hyperparameter tuning and has linear-in-$d$ complexity. We note that the matrix $X$ of neurons activated with the ReLu activation can be sparse. This implies that RandNets' cost is further improvable as the complexity of sparse operations is proportional to the number of nonzero matrix entries.
> - _Complex geometries in higher dimensions_: _F_ can be a symplectic, variational, energy-preserving, or any other integrator preserving the system's geometric properties. Using RandNet-Parareal with such _F_ is straightforward.
>
> **W4 & Q4: Discussion of poor scaling of RandNets to deep architectures and possible gains of using a deep network instead of a shallow one**
> **A**: The poor scaling of RandNets to deep architectures is not always observed. Statistical properties of RandNets (or random feature NNs) have been studied in a sizeable body of works [18,19]. Asymptotic characterizations for the test error of _shallow_ RandNets have been derived [15,16,17]. Similar studies in the deep case [13,14] show the equivalence of deep RandNets to deep linear Gaussian models and, importantly, provide instances where testing performance _improves_ as a function of depth. Nevertheless, deep RandNets require higher training cost, so using _shallow_ RandNets may still be preferable in our method. We explore this in our ongoing work.

---

> > ### Comment · Area_Chair_ACQd · 2024-08-13
> > **Last day for discussion**
> >
> > Reviewer nFhC, today is the last day for discussion. I hope you can take a moment to respond to the authors' rebuttal.

---

> > ### Comment · Reviewer_nFhC · 2024-08-13
> > **Raising my score**
> >
> > Thank you so much for providing detailed answers. Considering all the answers, I have raised my score from 5 to 6. It would be nice to include trade off results in the revised paper.

---

### Official Review · Reviewer_Eu5i · 2024-07-08

**Soundness:** 3
**Presentation:** 3
**Contribution:** 3
**Rating:** 7
**Confidence:** 4

**Summary:**

The authors introduce a numerical algorithm that computes the solution to a large system of ordinary differential equations (ODE) "parallel in time". The main idea of the solver is based on the existing "Parareal", which introduces parallelism by running a sequential, fast, and inaccurate ODE solver and then corrects it in parallel with a more accurate correction scheme. The novelty in this manuscript is that this correction scheme is done using random neural networks, i.e., networks where the hidden weights and biases are chosen at random and then fixed. The benefit of this is that training time for the network is reduced significantly, as only the last layer must be approximated, which is possible using a linear solve. The authors demonstrate this efficiency and the parallel in time property on several nonlinear, time-dependent partial differential equations, discretized to obtain the required large ODE system.

**Strengths:**

Using random neural networks in a parallel-in-time setting is (to my knowledge) a new approach, and helps to mitigate issues with earlier versions of algorithms (e.g. the GP version of ParaReal, as presented). To that end, the work is a combination of well-known techniques to solve a problem in a new way. The authors discuss and demonstrate how their approach differs from earlier work, mostly from ParaReal with GP, fine-scale solvers, and a classical ParaReal approach. The paper is very well written, with clear descriptions of the algorithm and numerical results, and even includes scaling results for multi-core systems. The authors also include a robustness study of the approach in the appendix.

**Weaknesses:**

1) There is no theoretical analysis beyond restating existing results from the literature. For example, there is no complexity analysis for the random network setting, which should have been relatively straightforward (given that it involves only a single linear solve, for which complexity results are available).
2) The numerical scenarios chosen in the paper are not particularly challenging on their own. It seems to me that the parallel in time setting itself is what makes them challenging, not the high number of degrees of freedom (e.g., 10^5 is not very large), nor the particularly high spatial dimension (one and two), or the complexity of the PDE (mildly nonlinear). Of course, for parallel-in-time algorithms these are challenging problems nonetheless, but the examples do not demonstrate why parallel in time in general makes more sense than just solving the system with a better algorithm (e.g. higher order, other basis functions, etc.). For example: it is not reasonable that solving Burgers equation on a 1D line takes 13 hours on the "fine scale" (table 1). This looks like an unnecessary difficulty for a problem that should be simple to solve, especially with only 128 degrees of freedom (variable d).
3) It seems to me that the main novelty of this paper is to replace the (numerically suboptimal) Gaussian process with a neural network where the internal weights are randomly chosen. It is not clear to me why this particular change is any better than improving the numerics for the GP setting, or using any other possible solution (e.g., just a linear mapping as approximation, or running a fine-scale solver on multiple scales, or using polynomial regression, ...). It is probably not possible to perform these additional experiments during the rebuttal time, but the manuscript does not even state any other possibilities that were ruled out.
4) Similarly, the state of the art focuses very much on ParaReal, and not on any other parallel in time solver. While this may be acceptable if ParaReal was the most advanced PinT solver available, it is not clear (because it is not stated) if that is the case in general (or at least for the chosen examples).

A few minor issues:
1) I would avoid using "nns" for nearest neighbors (l.35), because it is confusing with "NNs" (neural networks).
2) l.87: it is strange to say "24 core processors, 48 cores". It is clear that 24*2=48; I hope that is what the authors mean.

**Questions:**

1) The authors stress that training a GP is fundamentally more expensive than training the random network. However, I fail to see this: if we do not compute a full kernel matrix of N by N (N being the number of data points), but just randomly choose M center points and then solve a linear system of N by M, the complexity is exactly the same as in the random network case. The only difference is that instead of M neurons we have M kernel functions (k(x_i, *)), essentially forming a "radial basis neural network". This has been discussed at length in the GP literature, and many methods for this exist (e.g. Nystroem kernel approximations, inducing points methods, etc.). Why do the authors only compare to the full kernel case, which obviously is subpar in performance?

2) Why do the authors compare to clearly sub-optimal numerical solvers in the PDE setting? I may have missed the reasoning why even the simplest PDE settings take so much time to solve.

3) Is Parareal the only parallel-in-time solver available? The paper focuses mostly on different Parareal settings and approaches, but not on different parallel-in-time solution methods.

4) How does the high ambient-space dimension d of the state U affect the nearest neighbor search?

**Limitations:**

The authors are missing a discussion that "nearest neighbor" algorithms do not work well in high-dimensional settings (as given here, with $d \gg 1000$ in many cases). They also do not discuss the suboptimal scaling of the linear solution for the random neural networks (caused by the linear solve necessary for the outer weights).

---

> ### Author Rebuttal · Authors · 2024-08-07
>
> We thank you for the critical assessment of our work. We identified these main criticisms: (a) lack of a computational complexity study; (b) comparison to a GP with full kernel and lack of alternatives to the GP solution; \(c\) use of a suboptimal fine solver _F_. We addressed (a) conducting a detailed complexity study. We believe that (b)-\(c\) are based on a misconception and provide our explanation together with answers to questions (Q), weaknesses (W), and limitations (L). We hope it is evident that the points raised do not warrant a rejection of our work.
>
> **W1: Lack of complexity analysis**
> - The complexity of RandNet-Parareal is reported in Tbl. A [pdf] and is linear in $d$, while nnGParareal scales as $d^2$.
> - The theoretical computational costs as a function of $d$ and $N$ are plotted in Fig.A-B [pdf], confirming the empirical superior scalability of RandNet-Parareal given in Fig.1 [paper].
> - The speed-up of each approach can be immediately obtained as the ratio of the sequential runtimes of _F_ and the parallel runtimes, as in [1].
>
> **Q1: The proposed approach is only compared to a GP based on full-rank matrix**
> This statement is not correct. The competing approach does not use the GP with the full kernel matrix of $N\times N$ ($N$ is the sample size), but only its reduced $m \times m$ version with $m \ll N$ nearest neighbors (nnGP [1]). This is the only existing PinT scheme tackling the full GP kernel issue.
>
> **Q2 & W2: Choice of subobtimal numerical solvers and high running times**
> Although valuable for sequential solvers, these comments are not pertinent for PinT schemes. We will add the following comment to a camera-ready version for a better understanding of the customary empirical design for PinT schemes.
> - *Goal of PinT schemes:* There is some confusion regarding this point. The essence of any Parareal method is to obtain in a PinT manner the $\epsilon$-close solution to that produced sequentially by a generic _F_. Each Parareal variant is agnostic to the choice of _F_, and "canonical" off-the-shelf numerical solvers are standard in the PinT literature (here, Runge-Kutta, as in [1]). WLOG, more appropriate (accurate/faster) _F_ can be used. This does not contrast our framework: RandNet-Parareal allows the seamless replacement of _F_ and inherits its properties.
> - *Contribution and choice of slow _F_*: As we propose a new PinT scheme, we compare it with existing PinT approaches rather than ad-hoc sequential solvers, focusing on speed-up rather than absolute performance of _F_. Burgers' PDE is particularly slow as we used the same high accuracy setup of [1].
> - *Additional challenging example*: Despite the short rebuttal time, we ran 2 more challenging examples (2D and 3D Brusselator PDE) with results in Fig.C [pdf]. Should we add them to Suppl. Material?
>
> **W3: Choice of RandNet instead of better numerics for the GP/alternative solutions**
> A recent work [1] replaced the numerically suboptimal GP from [3] with nearest neighbor GPs. While improving GP numerics, it still suffers from quadratic complexity in $d$ (see Q1) and is sensitive to hyperparameter tuning (Tbl. A [pdf]). We propose abandoning the GP framework and using RandNets.
>
> Alternative choices, e.g. linear/polynomial functions, do not offer the desirable learning quality:
>  - *Universal approximation (UA) properties:* Linear functions do not form a UA class, so do not guarantee high-accuracy learning. Polynomials are UA, but (1) their order depends on the problem; (2) they are known to yield ill-conditioning in the regression setting. Instead, RandNets (as neural network functions) are UA [4] and do not suffer from (1)-(2).
> - *Overcoming the curse of dimensionality (CoD):* Under specific assumptions, one can determine if RandNets suffer from the CoD [4] (see also Q3 of Ref nFhC). This opens multiple avenues for provable speed and uncertainty quantification improvements with RandNet-Parareal.
>
> Despite limited rebuttal time, we tested $p$-order polynomials on Diffusion-Reaction PDE, finding they lack sufficient learning accuracy ($p=1$ is linear). The table displays iterations to convergence $K$ and parallel speed-up (in parentheses). For all models $K=N=64$ (so they converge serially), while for RandNet-Parareal, $K=12$ with a speed-up of 5.36 times (Tbl. 6 [paper]).
> |PDE|$p=1$|$p=2$|$p=3$|$p=5$|$p=7$|
> |-|-|-|-|-|-|
> |$d=722, N=64$|64 (0.98)|64 (0.95)|64 (0.92)|64 (0.88)|64 (0.83)|
>
> **W4 & Q3: Is Parareal the state-of-art among PinT schemes?**
> Yes. Due to space constraints, we kindly refer you to our detailed answer to Q1 of Ref FREq who posed a similar question. We will add the needed background to the revised introduction.
>
> **W5: nns (nearest neighbors) vs NNs (neural networks)**
> This may indeed be confusing. In the literature, nearest neighbor GPs are typically denoted as nnGPs, while neural networks as NNs. We welcome alternative suggestions.
>
> **W6: 24 core processors, 48 cores**
> In the updated version, we will change this to read as " 24 core processors, _yielding a total of_ 48 cores."
>
> **L1 & Q4: Issues of nearest neighbor algorithms in high dimensions**
> - *Suboptimal scaling of the linear solution for RandNets:* As shown in Tbl. A [pdf], it contributes as $M^3$ ($M$ the number of neurons) and has limited effect on performance for sensible $M$ values for our examples of (O/P)DE, spatial dimension, degrees of freedom.
> - *Complexity in high dimensions:* Naive linear search of the nearest neighbor is linear in $N$ and $d$, $O(Nd)$, which we took into account in complexity in WC1. Approximate Near. Neighbors would allow for further improvements [2].
> - *Performance in high dimensions:* Fig.D [pdf] shows that nearest neighbors (closest points in L2 distance) of target points are recovered for 2 high-dimensional systems $d>1e^4$, proving the validity of our approach. This nn structure is a feature of Parareal -- the dataset is _not made of random_ observations but of initial conditions converging in $k$.

---

> > ### Comment · Reviewer_Eu5i · 2024-08-10
> >
> > I highly appreciate the additional results, both experimental and theoretical. I very much like the general idea of the paper but was unsure about the soundness in these two aspects, both of which were addressed. I will raise my score to 7.
> > * "Should we add [the new experiments] to Suppl. Material?" Yes, please.

---

> > > ### Author Response · Authors · 2024-08-11
> > > **Follow-up to Reviewer Eu5i's comment**
> > >
> > > We are glad you appreciated our additional theoretical and numerical results. We thank you again for your constructive criticisms, we believe the revised paper will greatly benefit from them. We are also extremely grateful that you have upgraded your score notably.

---

### Official Review · Reviewer_FREq · 2024-07-12

**Soundness:** 3
**Presentation:** 2
**Contribution:** 2
**Rating:** 5
**Confidence:** 2

**Summary:**

The paper introduces their method RandNet-Parareal, which is a method to solve differential equations. Their method can be categorized as a Parallel-in-time technique that aims at parallelizing solvers in the temporal domain. They extend the Parareal algorithm by using RandNets which learn the difference between a coarse and a fine solver. They show experimentally a speed-up of their method compared to previous methods and provide theoretical guarantees for their method.

**Strengths:**

The paper evaluates their method on three different PDEs.
The authors show that their method yields a significant speed-up.
The authors provide theoretical guarantees of their method.
The authors claim that the algorithm can be used as a convenient out-of-the-box algorithm.

**Weaknesses:**

Paper Structure:
The authors provide experimental details within the introduction (lines 85-90) that should belong to the experimental section.
The paper is not structured as most papers in the conference. I think Section 2 should be put into a Section called “Problem Description”. Section 3 should be included in Section 2 or into a “Related Work” section. The paper misses a related work section.

To me, it is questionable if this paper is of interest to the audience of the conference. The paper replaces a subpart of a numerical algorithm with a two-layer Neural Network where only the last layer is trained. Therefore, while the contribution is sound the impact of the work to other fields of machine learning is very limited.

**Questions:**

1. Can you put your paper into a broader perspective? This paper only compares methods that are based on the Parareal algorithm. Are there also alternatives and how do they compare to your method?
2. What is the “Fine” algorithm in Table 1 and in Fig. 1? I could not find its definition.
3. How can the Runtime in Figure 1 be negative? Is log(Runtime) shown in this plot?
4. Is there a way to measure the solution quality of the solver? Is the solution quality fully defined by the converged initial condition $\epsilon$ and therefore the same for all methods?
If the solution quality can be quantified, how does the solution quality of your method compare to other methods?
5. Are the subscripts $k$ and $i$ dropped in Equation 6? If so this is a bit confusing since these are used in the previous Section and then dropped without a notice.

**Limitations:**

The authors addressed some limitations of their work.

---

> ### Author Rebuttal · Authors · 2024-08-07
>
> We appreciate that you recognize how the proposed method yields a significant speedup compared to competing ones. We believe to have successfully replied and tackled all your questions (Q) and weaknesses comments (W). We hope that you reconsider your score based on the new provided information.
>
>
> **Q1: Lack of background on existing parallel-in-time (PinT) schemes**
> **A**: This is a very important point, which we will address by adding the relevant background in the revised introduction. There are three general approaches for PinT computation: parallel across-the-problem, parallel-across-the-step, and parallel-across-the-method. [8,11] provide another classification: multiple shooting, methods based on waveform relaxation and domain decomposition, multigrid approaches, and direct time-parallel methods. Parallel-across-the-step methods, in which solutions at multiple time-grid points are computed simultaneously, include the Parareal (approximation of the derivative in the shooting method), Parallel Full Approximation Scheme in Space and Time (PFASST) (multigrid method) [5,10], and Multigrid Reduction in Time (MGRIT) [6,7] methods (see [9] for details).
>
> Among them, Parareal has received more attention in the literature, with extensive theoretical analyses, improved versions, and empirical applications [8,11]. Limited theoretical results are available for MGRIT and PFASST, with a few extensions and empirical applications. Interestingly, combined analyses have shown equivalences between Parareal and MGRIT, and connections between MGRIT and PFASST.
>
> It is well acknowledged that comparing PinT methods based on different working principles is extremely hard, with [11] representing a recent survey article which nonetheless does not offer a systematic comparison either. Quoting [11], "caution should be taken when directly comparing speedup numbers across methods and implementations. In particular, some of the speedup and efficiency numbers are only theoretical in nature, and many of the parallel time methods do not address the storage or communication overhead of the parallel time integrator". [9] is one of very few recent attempts to systematically compare different PinT classes. However, it is limited exclusively to the Dahlquist problem. Thus, it has become conventional to compare new techniques to the existing state-of-the-art methods within the same group of solvers. This is what we do in our paper, comparing RandNet-Parareal with the original Parareal and the recently improved version nnGParareal [1]. A broader comparison with alternative PinT approaches would be insightful, but it is beyond the scope of this work.
>
>  **Q2: What is the “Fine” algorithm in Table 1 and in Fig. 1?**
> **A**: The fine solver $\mathcal{F}$ is the accurate solver defined on Page 1, line 28 in our submission. $\mathcal{F}$ is typically chosen to be a higher accuracy method (e.g. Runge-Kutta 8) compared to the coarse solver (e.g. Runge-Kutta 2).
>
> **Q3: How can the Runtime in Figure 1 be negative? Is log(Runtime) shown in this plot?**
> **A**: Yes, it is the $\log_{10}$ runtime, as specified in the y-axis on Fig. 1 [paper].
>
> **Q4: Is there a way to measure the solution quality of the solver? How does the solution quality of the method compare to other methods?**
> **A**: The fine solver $\mathcal{F}$ is chosen by the user, and it determines the accuracy of the solution. All Parareal-type PinT schemes target such solution, with closeness controlled by $\epsilon$, i.e., all converged solutions will be $\epsilon$-close to the one of $\mathcal{F}$. In the table below, we report the maximum absolute error committed by each of the RandNet-Parareal, Parareal, and nnGParareal with respect to the fine solver (run sequentially), averaged over intervals, together with the runtime in parentheses.
>
> |PDE|RandNet-Parareal|Parareal|nnGParareal|
>  | - | - | - | - |
>  |Burgers' $d=128$| $1.06e^{-8}$ (1h 2m)|$1.85e^{-8}$ (8h 54m)|$1.32e^{-7}$ (1h 39m)|
>  |Diffusion-Reaction $d=7.2e^2$|$3.56e^{-8}$ (23m)|$1.83e^{-8}$ (1h 40m)|$5.71e^{-7}$ (1h 11m)|
>  |Diffusion-Reaction $d=3.3e^3$|$8.56e^{-10}$ (33m)|$2.45e^{-8}$ (7h 52m)|not converged|
>  |Diffusion-Reaction $d=2.5e^4$|$8.09e^{-11}$ (1h 57m) |$7.43e^{-9}$ (9h 50m)|not converged|
>  |SWE $d=3.1e^4$|$6.75e^{-8}$ (4h 9m)|$5.15e^{-8}$ (15h 43m)|not converged|
>  |SWE $d=6.1e^4$|$8.54e^{-9}$ (12h 34m)|$2.84e^{-8}$ (19h 30m)|not converged|
>
> **Q5: Are the subscripts $k$ and $i$ dropped in Eq. 6?**
> **A**: Yes, they are indeed dropped, since we first present our method for a generic setting, and then use it for the specific inputs, adding the indices again on lines 237-238.
>
> **W1: Paper structure**
> We did not find any specific guidelines in terms of paper structure, and we have seen numerous papers with similar structure to ours. Note that what you call "Related work" is in fact Section 3, where we introduce two recent works improving on Parareal. We will nevertheless take your comments on board when preparing the camera-ready version.
>
> **W2: Lack of interest for the audience of the conference**
> We disagree with you, as this is not the first contribution on PinT methods appearing on NeurIPS. You write that *"while the contribution is sound, the impact of the work to other fields of machine learning is very limited"*. However, this is not a NeurIPS requirement, as long as the paper matches the subjects and topicality of the Primary Area. Nevertheless, we believe that our new randomized PinT numerical solver has far-reaching implications for the scientific computing field and others, where efficient numerical (O/P)DEs solvers are in need. We kindly refer you to our detailed answer to W1 of Ref awZW. There, we discuss the importance of PinT schemes for scientific machine learning (ML)/computing and science in general and the use of scientific PinT computing in other fields of machine learning (e.g. diffusion models, neural ODEs, optimization, normalizing flows, to name a few).

---

> > ### Comment · Reviewer_FREq · 2024-08-08
> > **Answer to Rebuttal**
> >
> > **AQ1:**
> > I appreciate the authors clarification on the background and encourage the authors to add a corresponding section into the paper.
> >
> > **AQ2:**
> > I think this should be defined more clearly in the experiment section.
> >
> > **AQ3:** The y-axis label is incorrect, as it is denoted in hours but should be log10(hours). It would be better to scale the y-axis logarithmically while keeping the y-axis ticks in hours, as this is easier to interpret than log10(hours).
> >
> > **AQ4:** The authors should not drop indices without mentioning it, and they should add a footnote for clarification.
> >
> > **AW2:** While this may not be a NeurIPS requirement, the reviewers should still consider the potential impact of the method in AI when assigning ratings. The reviewers noted the method's potential applicability to ODE solvers used in machine learning, such as in diffusion models. The paper would benefit from a section discussing this potential application in more detail.
> >
> > **Further Question:** Are PinT schemes used to solve the reverse ODE in Diffusion Models? If not, which ODE solvers are used? What would be necessary to apply the authors' method in this setting? Likely it would need to be implemented on a GPU?
> >  Demonstrating the benefit of the authors' method in the context of diffusion models would significantly increase the relevance of this paper to the NeurIPS community.

---

> > > ### Author Response · Authors · 2024-08-09
> > > **Follow-up answer to Reviewer FREq's comments**
> > >
> > > **AQ1-3**: Thank you for your feedback. We will carefully take it into account when preparing the final version.
> > >
> > > We welcome any comment on our response to **Q4** on the solution quality of the solver.
> > >
> > > **AQ5: Dropping indices without prior mention.**
> > > Please note that we specify that on Line 195 in [paper], which reads "Prior to that [i.e., defining RandNet for Parareal], we define how RandNets work in a general setting with input $\boldsymbol{U}$". Indices are introduced later, upon combining RandNets with Parareal, in line 237. Should you find this still unclear to the reader, we could change it to "Prior to that, we define how RandNets work in a general setting with input $\boldsymbol{U}$, *before going back to the input of interest $\boldsymbol{U}_i^k$ within the Parareal framework*".
> > >
> > > **AW2 and Further Q: On the potential impact of RandNet-Parareal on ML/AI, and, specifically, on Diffusion Models (DMs).**
> > > Thank you for engaging in further conversation about this matter. Properly communicating the relevance and impact of our method is very important for us.
> > >
> > > Our work exemplifies an instance when ML methods allow the advancement of various fields of science and engineering where solving (O/P)DEs is needed, see also our response to W2 above, and W1 of Ref awZW. We mentioned these applications as our submission belongs to the Primary Area of *ML for Physical Sciences*.
> > >
> > > However, AI is also one of the fields that could greatly benefit from progresses regarding efficient ODE solvers, and thus our method. This is why we gladly provide examples of how our approach can also assist ML and AI, adding these points in the camera-ready version of the paper.
> > >
> > > ODEs are a crucial building block of some relevant techniques, such as *DMs*, *Neural ODEs*, *Optimal control and reinforcement learning*, and *Optimization for ML models*, to name a few.
> > >
> > > - _Solvers for DMs_: In the context of DMs, mentioned in your answer, the continuous time reversal process can be described by a probability flow ODE, defined by the score function, usually approximated with deep (convolutional) NNs (note that the ODE formulation offers significant advantages over SDEs in high dimensions). Faster ODE solvers can improve sampling speed, yielding faster image synthesis. The main existing directions in the literature involve using classical sequential solvers [30], developing faster and dedicated ones such as DDIM [24], DDPM [29], DPM-Solver [25], Heun [26], and parallelization of the autoregressive sampling process of DMs (by using Picard-Lindelöf iterations for ODE/SDE) [27,28]. The latter emerged mainly due to the need for other solutions to avoid the bottlenecks of sequential solvers. To the best of our knowledge, modulo this particular parallelized sampling, PinT schemes have not yet been used in this context. **This is the gap this paper could fill, as the existing successful sequential dedicated solvers could be embedded into our proposed RandNet-Parareal**. We expect this to be straightforward, since the goal is to collect samples via solving the corresponding (diffusion) ODEs at $[0,T]$, where RandNet-Parareal immediately finds its purpose.
> > >
> > > - _GPU Implementation_: There are two increasingly sophisticated approaches for implementing RandNets-Parareal on GPUs: (i) parallel implementation of the solvers (model parallelism); (ii) parallel implementation of Parareal (in-time parallelism). For (i), both the solver and the RandNet computation can be carried out on the GPU, see e.g.[31]. In-time parallelism would then be implemented across GPUs. Otherwise, under (ii), both model and time parallelism can be implemented concurrently on the same hardware [32]. The latter guarantees a better use of resources at the cost of increased implementation complexity.
> > >
> > > We hope these clarifications provide you with evidence of the expected impact/implications of our method in the context of generative AI models and beyond.
> > >
> > > **References used above:**
> > > [24] J. Song, C. Meng, S. Ermon. Denoising diffusion implicit models. 2020.
> > > [25] C. Lu, Y. Zhou, F. Bao, J. Chen, C. Li, J. Zhu. DPM-solver: A fast ODE solver for diffusion probabilistic model sampling in around 10 steps. 2022.
> > > [26] T. Karras, M. Aittala, T. Aila, S. Laine. Elucidating the design space of diffusion-based generative models. 2022.
> > > [27] A. Shih, S. Belkhale, S. Ermon, D. Sadigh, N. Anari. Parallel sampling of diffusion models. 2023.
> > > [28] Z. Tang, J. Tang, H. Luo, F. Wang, T.-H. Chang. Accelerating parallel sampling of diffusion models. 2024.
> > > [29] J. Ho, A. Jain, P. Abbeel. Denoising diffusion probabilistic models. 2020.
> > > [30] Y. Song, J. Sohl-Dickstein, D.P. Kingma, A. Kumar, S. Ermon, B. Poole. Score-based generative modeling through stochastic differential equations. 2021.
> > > [31] A.N.Budko, M. Möller, C. W. J. Lemmens. Time integration parallel in time. 2020.
> > > [32] A.Q. Ibrahim, S. Götschel, D. Ruprecht. Parareal with a physics-informed neural network as coarse propagator.

---

> > > > ### Comment · Reviewer_FREq · 2024-08-12
> > > > **Raised Score**
> > > >
> > > > I appreciate the authors' efforts in addressing the concerns raised in the rebuttal. Based on this, I am increasing my score from 4 to 5.
> > > >
> > > > The paper presents an improvement to a numerical ODE solver, which is solely relevant to obtaining training data for PDE machine learning methods.
> > > >
> > > > To enhance the paper's relevance to the AI community, I suggest two improvements:
> > > >
> > > > 1.  Provide a GPU implementation of the proposed method.
> > > > 2.  Include a comparison between this method and other ODE solvers commonly used in simulating the reverse process of diffusion models.
> > > >
> > > > These additions would help demonstrate the practical applications and potential advantages of the method in AI-related contexts.

---

> ### Author Response · Authors · 2024-08-12
> **Follow-up answer to Reviewer FREq's comments and raised score**
>
> We are glad that our rebuttal comments were helpful in addressing your concerns. We are also grateful for reflecting this in your score, and for your further suggestions.
>
> We notice that there are still some lingering misconceptions about our work, which we believe are important to point out.
>
> Our work does not "present an improvement to a numerical ODE solver". Instead, it improves existing _Parallel-in-Time_ (PinT) methods for solving ODEs and PDEs, as written in the paper abstract, summarized by the other reviewers, and discussed at length during the rebuttal. We do, ultimately, solve an (O/P)DE with a numerical scheme, but our proposed scheme
> (1) solves DEs by parallelizing in time, belonging thus to a different class than that of the "simple" numerical ODE solvers;
> (2) models the correction term using particularly suited **random neural networks**, allowing us to achieve significant speed-up compared to existing PinT techniques;
> (3) can incorporate existing (O/P)DE solvers in its architecture;
> (4) is not meant to improve a specific ODE solver, but instead to make PinT techniques more effective and, hence, more appealing to practitioners in the field.
>
> Moreover, our new approach (even if considered to be an improvement to an ODE solver, an oversimplification which we disagree with) is far from "solely relevant to obtaining training data for PDE machine learning methods". Numerical methods for solving DEs have profoundly impacted scientific progress since the last century, extending far beyond the generation of training data generation. In fact, once again, **our method can incorporate existing PDE machine learning solvers instead of training data for them.** Independently on that, landmark achievements such as weather forecasting, space exploration, and advancements in nuclear energy are just a few examples of the broad and transformative influence of numerical DE solvers.
>
> Commenting on the relevance of designing efficient PinT solvers beyond AI, the following non-exhaustive list contains some other areas of natural sciences, social sciences, and engineering where the availability of efficient (O/P)DE solvers could be a game-changer leading to extremely important advances:
>
> - *Atmospheric Dynamics:* Accurate solutions can mitigate climate change, enhance disaster preparedness, and optimize agriculture.
> - *Biomedical Processes:* Real-time simulations of biological processes, such as tumor growth and blood flow, can aid in personalized treatment planning and targeted interventions.
> - *Earthquake Engineering:* Refined modeling of seismic wave propagation and structural responses enhances earthquake predictions and leads to more resilient building designs.
> - *Financial Modeling:* Improved pricing accuracy and real-time risk assessment can stabilize financial markets and reduce systemic risk.
>
> We believe our work is another step toward providing faster numerical solutions for DEs in any field where this may be required, whether in the natural sciences, engineering, machine learning, or artificial intelligence.
>
> We thank you for raising the score and hope that our further comments help convey the main scope of our contributions.

---

### Author Rebuttal · Authors · 2024-08-07

We would like to thank the reviewers for their constructive feedback and suggestions. We are happy to read that the paper "is well-written and well-organized" (awZW), "very well-written" (Eu5i), "contains clear descriptions and robustness study" (Eu5i) and that the proposed method is a "new approach" (Eu5i) with "provided theoretical guarantees" (FREq) which "yields a significant speed-up" compared to previous methods (FREq) and "helps to mitigate issues with earlier versions of algorithms" (Eu5i). We are particularly encouraged by the observation that "considered PDEs are ... complicated and exhibit the efficacy of the proposed approach" (nFhC). We are glad that all reviewers agree on our contribution being the first one that embeds NNs within parallel-in-time (PinT) schemes. We hope this work will prompt further research in this area, ideally combining the recent advancements in neural PDEs solvers with PinT schemes.

We are grateful to the reviewers for suggesting several improvements, which will notably enhance the revised version of the paper. Two common objections are (i) the lack of background on alternative parallel-in-time (PinT) schemes and (ii) a discussion on the computational complexity of the proposed approach (with respect to both competing methods and accuracy). We addressed point (i) in the rebuttal comments, while for (ii) we derived the theoretical complexity of our method and compared it to the recently introduced nnGParareal (see Table A in pdf).

Additionally, during the rebuttal period, following suggestions of Eu5i, we tested our method on two new challenging examples, the 2D and 3D Brusselator PDEs, which are known to exhibit complex behavior, including oscillations, spatial patterns, and chaos. We also compare RandNets with polynomial functions, showing the latter's suboptimal performance.

The attached pdf (cited as [pdf]) includes additional figures and the complexity table. In the camera-ready version, we will place our contribution within existing PiT schemes, add a discussion on computational complexity, and include results obtained during the rebuttal time in Supplementary Material if the reviewers deem it advisable.

We believe that our new randomized PinT numerical solver has far-reaching implications for the scientific computing field specifically and for science/engineering more broadly, where time- and space-efficient solving of (O/P)DEs remains one of the most important research directions. By combining randomized neural networks with existing PinT methodology, we provide a framework that couples theoretical and robustness guarantees with a drastic reduction of computational requirements, as demonstrated through extensive simulation. Finally, our method can offer speedups for numerous instances where solving ODEs in the machine learning realm is needed (for example, in the context of diffusion models, neural ODEs, and many others).

We are deeply grateful for the helpful feedback received from the reviewers, which will make the final version of our manuscript (especially its introduction) more comprehensive and refined.

**References used in the rebuttals:**

[1] G. Gattiglio, L. Grigoryeva, M. Tamborrino. Nearest neighbors GParareal: Improving scalability of Gaussian processes for parallel-in-time solvers. 2024.
[2] P. Indyk, R. Motwani. Approximate nearest neighbors: towards removing the curse of dimensionality. 1998.
[3] K. Pentland, M. Tamborrino, T. Sullivan, J. Buchanan, L. Appel. GParareal: a time-parallel ODE solver using Gaussian process emulation. 2023.
[4] L. Gonon, L. Grigoryeva, J.-P. Ortega. Approximation bounds for random neural networks and reservoir systems. 2023.
[5] M. Emmett, M. Minion. Toward an efficient parallel in time method for partial differential equations. 2012.
[6] R. Falgout, S. Friedhoff, T. Kolev, S. MacLachlan, J. Schroder. Parallel time integration with multigrid. 2014.
[7] S. Friedhoff, R. Falgout, T. Kolev, S. MacLachlan,  J. Schroder. A multigrid-in-time algorithm for solving evolution equations in parallel. 2012.
[8] M. Gander. 50 years of time parallel time integration. 2015.
[9] M. Gander, T. Lunet, D. Ruprecht, R. Speck. A unified analysis framework for iterative parallel-in-time algorithms. 2023.
[10] M. Minion. A hybrid parareal spectral deferred corrections method. 2010.
[11] B. Ong, J. Schroder. Applications of time parallelization. 2020.
[12] A. Neufeld, P. Schmocker. Universal approximation property of random neural networks. 2023.
[13] D. Schröder, H. Cui, D. Dmitriev, B. Loureiro. Deterministic equivalent and error universality of deep random features learning. 2023.
[14] D. Bosch, A. Panahi, B. Hassibi. Precise asymptotic analysis of deep random feature models. 2023.
[15] S. Mei, A. Montanari. The generalization error of random features regression: Precise asymptotics and the double descent curve. 2022.
[16] H. Hu, Y. Lu. Universality laws for high-dimensional learning with random features. 2022.
[17] F. Gerace, B. Loureiro, F. Krzakala, M. Mezard, L. Zdeborova. Generalisation error in learning with random features and the hidden manifold model. 2020.
[18] J. Lee, Y. Bahri, R. Novak, S. Schoenholz, J. Pennington, J. Sohl-Dickstein. Deep neural networks as Gaussian processes. 2018.
[19] A. De G. Matthews, J. Hron, M. Rowland, R. Turner, Z. Ghahramani. Gaussian process behaviour in wide deep neural networks. 2018.
[20] F. Hamon, M. Schreiber, M. Minion. Parallel-in-time multi-level integration of the shallow-water equations on the rotating sphere. 2020.
[21] D. Samaddar, D. Coster, X. Bonnin, L. Berry, W. Elwasif, D. Batchelor. Application of the parareal algorithm to simulations of ELMs in ITER plasma. 2019.
[22] R. Chen, Y. Rubanova, J. Bettencourt, D. Duvenaud. Neural ordinary differential equations. 2018.
[23] G. Papamakarios, E. Nalisnick, D. Rezende, S. Mohamed, B. Lakshminarayanan. Normalizing flows for probabilistic modeling and inference. 2021.

---

### Decision · Program_Chairs · 2024-09-25

**Decision:**

Accept (poster)

**Comment:**

This paper continues a line of work on *Parareal methods for accelerating PDE solvers on parallel machines. The results demonstrate significant speedup on a variety of canonical problems. The innovation is somewhat modest, employing well-studied random neural network as a drop-in replacement for Gaussian processes used in previous work.  Nevertheless, I think these results will be of interest to some in the community.